# Navigating to the Best Policy
# in Markov Decision Processes

**Aymen Al Marjani**
UMPA, ENS Lyon
Lyon, France
aymen.al_marjani@ens-lyon.fr

**Aurélien Garivier**
UMPA, CNRS, INRIA, ENS Lyon
Lyon, France
aurelien.garivier@ens-lyon.fr

**Alexandre Proutiere**
EECS and Digital Futures
KTH Royal Institute of Technology, Sweden
alepro@kth.se

## Abstract

We investigate the classical active pure exploration problem in Markov Decision Processes, where the agent sequentially selects actions and, from the resulting system trajectory, aims at identifying the best policy as fast as possible. We propose a problem-dependent lower bound on the average number of steps required before a correct answer can be given with probability at least $1 - \delta$. We further provide the first algorithm with an instance-specific sample complexity in this setting. This algorithm addresses the general case of communicating MDPs; we also propose a variant with a reduced exploration rate (and hence faster convergence) under an additional ergodicity assumption. This work extends previous results relative to the *generative setting* [MP21], where the agent could at each step query the random outcome of any (state, action) pair. In contrast, we show here how to deal with the *navigation constraints*, induced by the *online setting*. Our analysis relies on an ergodic theorem for non-homogeneous Markov chains which we consider of wide interest in the analysis of Markov Decision Processes.

## 1 Introduction

Somewhat surprisingly, learning in a Markov Decision Process is most often considered under the performance criteria of *consistency* or *regret minimization* (see e.g. [SB18, Sze10, LS20] and references therein). Regret minimization (see e.g. [AJO09, FCG10]) is particularly relevant when the rewards accumulated during the learning phase are important. This is however not always the case: for example, when learning a game (whether Go, chess, Atari, or whatever), winning or losing during the learning phase does not really matter. One may intuitively think that sometimes getting into difficulty on purpose so as to observe unheard situations can significantly accelerate the learning process. Another example is the training of robot prototypes in the factory: a reasonably good policy is first searched, regardless of the losses incurred, that can serve as an initialization for a second phase regret-minimization mode that starts when the robot is deployed. It is hence also of great practical importance to study the sample complexity of learning, and to work on strategies that might improve, in this perspective, on regret minimizing algorithms.

In this work, we are interested in the best policy identification (BPI) problem for infinite-horizon discounted MDPs. This framework was introduced by [Fie94] under the name of *PAC-RL*. In BPI the algorithm explores the MDP until it has gathered enough samples to return an $\varepsilon$-optimal policy with probability at least $1 - \delta$. Crucially, the algorithm halts at a *random time step*, determined by a

35th Conference on Neural Information Processing Systems (NeurIPS 2021).

*stopping rule* which guarantees that the probability of returning a wrong answer is less that $\delta$. The optimality of BPI algorithms is measured through their *sample complexity*, defined as their expected stopping time. Best policy identification in MDPs has been mostly investigated under the lens of minimax-optimality. The minimax framework may be overly pessimistic by accounting for the worst possible MDP, whereas an algorithm with instance-specific guarantees can adapt its exploration procedure to the hardness of the MDP instance that it faces. Recently, a few works made the first attempts towards understanding the instance-specific sample complexity of reinforcement learning. However, they typically either make simplifying assumptions such as access to a generative model [ZKB19, MP21], or restrict their attention to episodic MDPs [WSJ21]. In practice however, the samples gathered rather correspond to a single, possibly infinite, trajectory of the system that we wish to control. This motivates our study of the full online setting where observations are only obtained by navigating in the MDP, that is by sequential choices of actions and following the transitions of the MDP.

**Our contributions.** [MP21] recently proposed an information-theoretical complexity analysis for MDPs in the case of access to a generative model. Here we extend their results to the online setting. Our main goal is to understand how the online learning scheme affects the sample complexity compared to the easier case where we have a generative model. A natural first step consists in understanding how the first order term $T^\star \log(1/\delta)$ changes. Thus we only focus on the asymptotic regime $\delta \to 0$. Our key contributions can be summarized as follows:

- First, we adapt the lower bound of [MP21] to the online setting (Proposition 2). The new bound also writes as the value of a zero-sum two-player game between nature and the algorithm, where the loss function remains the same, but where the set of possible strategies for the algorithm is restricted to a subset $\Omega(\mathcal{M})$ of the simplex of dimension $SA - 1$. We refer to the constraints defining $\Omega(\mathcal{M})$ as the *navigation constraints*.

- We propose MDP-NaS, the first algorithm for the online setting[1] with instance-dependent bounds on its sample complexity in the asymptotic regime $\delta \to 0$ (Theorem 6). A major challenge lies in the design of a sampling rule that guarantees that the sampling frequency of state-action pairs $\left(N_{sa}(t)/t\right)_{s,a}$ converges to some target oracle allocation $\omega^\star \in \Omega(\mathcal{M})$. Indeed, since we can no longer choose the next state of the agent, the tracking procedure which was developed by [GK16] for multi-armed bandits and used in [MP21] for MDPs with a generative model can no longer be applied in our setting. We propose a new sampling rule which performs exploration according to a mixture of the uniform policy and a plug-in estimate of the *oracle policy* (the policy whose stationary state-action distribution is $\omega^\star$) and prove that it satisfies the requirement above. The analysis of our sampling rule relies on an ergodic theorem for non-homogeneous Markov chains of independent interest (Proposition 12).

- We investigate, depending on the communication properties of the ground-truth instance $\mathcal{M}$, what is the minimal forced-exploration rate in our sampling rule that guarantees the consistency of the plug-in estimator of the oracle policy. Our findings imply that when $\mathcal{M}$ is ergodic, an exploration rate as low as $1/\sqrt{t}$ is sufficient. However, when $\mathcal{M}$ is only assumed to be communicating, one is obliged to resort to a much more conservative exploration rate of $t^{-\frac{1}{m+1}}$, where $m$ is a parameter defined in Lemma 4 that may scales as large as $S - 1$ in the worst case.

- Finally, our stopping rule represents the first implementation of the Generalized Likelihood Ratio test for MDPs. Notably, we circumvent the need to solve the max-min program of the lower bound exactly, and show how an upper bound of the best-response problem, such as the one derived in [MP21], can be used to perform a GLR test. This improves upon the sample complexity bound that one obtains using the KL-Ball stopping rule of [MP21] by at least a factor of $2$[2].

---

[1]Before publication of this work, but after a preprint was available online, [WSJ21] proposed another algorithm with instance-dependent guarantees.

[2]Note that the stopping rule is independent of the sampling rule and thus can be used in both the generative and the online settings. Furthermore, one may even obtain an improved factor of 4 by using a deviation inequality for the full distribution of (reward, next-state) instead of a union bound of deviation inequalities for each marginal distribution.

## 1.1 Related work

**Minimax Best-Policy Identification.** BPI in the online setting has been investigated recently by a number of works with minimax sample complexity bounds. In the case of episodic MDPs [KMDD+21], [MDJ+20] proposed algorithms that identify an $\varepsilon$-optimal policy at the initial state w.h.p. In contrast, in the case of infinite-horizon MDPs one is rather interested in finding a good policy at *every* state. Recent works provide convergence analysis for Q-learning [LWC+20b] or policy gradient algorithms [AHKS20], [FYAY21], [ZCA21]. Their results typically state that if the algorithm is fed with the appropriate hyperparameters, it can return an $\varepsilon$-optimal policy w.h.p. after collecting a polynomial number of samples. In practice, a pure exploration agent needs a stopping rule to determine when to halt the learning process. In particular, the question of how to tune the number of iterations of these algorithms without prior knowledge of the ground truth instance remains open.[3]

**Generative Model.** A large body of literature focuses on the so-called *generative model*, where in each step, the algorithm may query a sample (i.e., observe a reward and a next-state drawn from the rewards and transitions distributions respectively) from any given state-action pair [KS98],[KMN99], [EDMM06], [AMK13],[DTC13], [Wan17], [SWW+18], [ZKB19], [AKY20], [LWC+20a], [LCC+21], [MP21].

**Instance-specific bounds.** Instance-optimal algorithms for Best Arm Identification in multi armed bandits (MDPs with one state) have been obtained independently by [GK16],[Rus16]. Here, we extend their information-theoretical approach to the problem of Best Policy Identification in MDPs. More recently, [WSJ21] provided an algorithm for BPI in episodic MDPs with instance-specific sample complexity. A more detailed comparison with [WSJ21] can be found in Section 6.

**Outline.** The rest of the paper is organized as follows: After introducing the setting and giving some notation and definitions in Section 2, we derive in Section 3 a lower bound on the time required by any algorithm navigating the MDP until it is able to identify the best policy with probability at least $1 - \delta$. The algorithm is presented in Section 4. Section 5 contains our main results along with a sketch of the analysis. Finally, in Section 6 we compare our results with MOCA, the only other algorithm (to the best of our knowledge) with problem-dependent guarantees in the online setting. Most technical results and proofs are given in the appendix.

## 2 Setting and notation

**Discounted MDPs.** We consider infinite horizon MDPs with discount and finite state and action spaces $\mathcal{S}$ and $\mathcal{A}$. Such an MDP $\mathcal{M}$ is defined through its transition and reward distributions $p_{\mathcal{M}}(\cdot|s,a)$ and $q_{\mathcal{M}}(\cdot|s,a)$ for all $(s,a)$. For simplicity, $q_{\mathcal{M}}(\cdot|s,a)$ will denote the density of the reward distribution w.r.t. a positive measure $\lambda$ with support included in $[0,1]$. Specifically, $p_{\mathcal{M}}(s'|s,a)$ denotes the probability of transitioning to state $s'$ after playing action $a$ at state $s$ while $R(s,a)$ is the random instantaneous reward that is collected. Finally, $\gamma \in [0,1)$ is a discounting factor. We look for an optimal control policy $\pi^\star : \mathcal{S} \to \mathcal{A}$ maximizing the long-term discounted reward: $V_{\mathcal{M}}^\pi(s) = \mathbb{E}_{\mathcal{M},\pi}[\sum_{t=0}^\infty \gamma^t R(s_t, \pi(s_t))]$, where $\mathbb{E}_{\mathcal{M},\pi}$ denotes the expectation w.r.t the randomness in the rewards and the trajectory when the policy $\pi$ is used. Classically, we denote by $V_{\mathcal{M}}^\star$ the optimal value function of $\mathcal{M}$ and by $Q_{\mathcal{M}}^\star$ the optimal $Q$-value function of $\mathcal{M}$. $\Pi^\star(\mathcal{M}) = \{\pi : \mathcal{S} \to \mathcal{A}, V_{\mathcal{M}}^\pi = V_{\mathcal{M}}^\star\}$ denotes the set of optimal policies for $\mathcal{M}$.

**Problem-dependent quantities.** The sub-optimality gap of action $a$ in state $s$ is defined as $\Delta_{sa} = V_{\mathcal{M}}^\star(s) - Q_{\mathcal{M}}^\star(s,a)$. Let $\Delta_{\min} = \min_{s,a\neq\pi^\star(s)} \Delta_{sa}$ be the minimum gap and $\mathrm{sp}(V_{\mathcal{M}}^\star) = \max_{s,s'} |V_{\mathcal{M}}^\star(s) - V_{\mathcal{M}}^\star(s')|$ be the span of $V_{\mathcal{M}}^\star$. Finally, we introduce the variance of the value function in $(s,a)$ as $\mathrm{Var}_{p(s,a)}[V_{\mathcal{M}}^\star] = \mathrm{Var}_{s'\sim p_{\mathcal{M}}(.|s,a)}[V_{\mathcal{M}}^\star(s')]$.

**Active pure exploration with fixed confidence.** When $\mathcal{M}$ is unknown, we wish to devise a learning algorithm $\mathbb{A}$ identifying, from a single trajectory, an optimal policy as quickly as possible with some given level of confidence. Formally, such an algorithm consists of (i) a sampling rule, selecting in each round $t$ in an adaptive manner the action $a_t$ to be played; $a_t$ depends on past observations, it

---

[3]Here we chose not to mention works that investigate the sample complexity in the PAC-MDP framework [Kak03]. Indeed, in this framework the sample complexity is rather defined as the the number of episodes where the algorithm does not play an $\varepsilon$-optimal policy. As explained in [DMKV21], this objective is closer to regret minimization than to pure exploration.

is $\mathcal{F}_t^{\mathbb{A}}$-measurable where $\mathcal{F}_t^{\mathbb{A}} = \sigma(s_0, a_0, R_0 \ldots, s_{t-1}, a_{t-1}, R_{t-1}, s_t)$ is the $\sigma$-algebra generated by observations up to time $t$; (ii) a stopping rule $\tau_\delta$, a stopping time w.r.t. the filtration $(\mathcal{F}_t^{\mathbb{A}})_{t \geq 0}$, deciding when to stop collecting observations; (iii) a decision rule returning $\hat{\pi}_{\tau_\delta}^\star$ an estimated optimal policy.

An algorithm $\mathbb{A}$ is $\delta$-Probably Correct ($\delta$-PC) over some set $\mathbb{M}$ of MDPs, if for any $\mathcal{M} \in \mathbb{M}$, it returns (in finite time) an optimal policy with probability at least $1 - \delta$. In this paper, we aim to devise a $\delta$-PC algorithm $\mathbb{A}$ with minimal sample complexity $\mathbb{E}_{\mathcal{M}, \mathbb{A}}[\tau_\delta]$. We make the following assumption.

**Assumption 1.** *We consider the set $\mathbb{M}$ of communicating MDPs with a unique optimal policy.*

We justify the above assumption as follows. (i) We restrict our attention to the case where $\mathcal{M}$ is communicating, for otherwise, if it is Multichain there would be a non-zero probability that the algorithm enters a subclass of states from which there is no possible comeback. In this case it becomes impossible to identify the *global* optimal policy[4]. (ii) About the uniqueness of the optimal policy, treating the case of MDPs with multiple optimal policies, or that of $\varepsilon$-optimal policy identification, requires the use of more involved Overlapping Hypothesis tests, which is already challenging in multi-armed bandits (MDPs with a single state) [GK19]. We will analyze how to remove this assumption in future work.

**Notation.** $\mathcal{Z} \triangleq \mathcal{S} \times \mathcal{A}$ is the state-action space. $\Sigma \triangleq \{\omega \in \mathbb{R}_+^{S \times A} : \sum_{s,a} \omega_{sa} = 1\}$, the simplex of dimension $SA - 1$. $N_{sa}(t)$ denotes the number of times the state-action pair $(s, a)$ has been visited up to the end of round $t$. We also introduce $N_s(t) = \sum_{a \in \mathcal{A}} N_{sa}(t)$. Similarly, for a vector $\omega \in \Sigma$ we will denote $\omega_s \triangleq \sum_{a \in \mathcal{A}} \omega_{sa}$. For a matrix $A \in \mathbb{R}^{n \times m}$, the infinity norm is defined as $\|A\|_\infty \triangleq \max_{1 \leq i \leq n} \sum_{j=1}^m |a_{i,j}|$. The KL divergence between two probability distributions $P$ and $Q$ on some discrete space $\mathcal{S}$ is defined as: $KL(P\|Q) \triangleq \sum_{s \in \mathcal{S}} P(s) \log(\frac{P(s)}{Q(s)})$. For Bernoulli distributions of respective means $p$ and $q$, the KL divergence is denoted by $\mathrm{kl}(p, q)$. For distributions over $[0, 1]$ defined through their densities $p$ and $q$ w.r.t. some positive measure $\lambda$, the KL divergence is: $KL(p\|q) \triangleq \int_{-\infty}^\infty p(x) \log\left(\frac{p(x)}{q(x)}\right) \lambda(dx)$. $\mathbb{M}$ denotes the set of communicating MDPs with a unique optimal policy. For two MDPs $\mathcal{M}, \mathcal{M}' \in \mathbb{M}$, we say that $\mathcal{M} \ll \mathcal{M}'$ if for all $(s, a)$, $p_\mathcal{M}(\cdot|s, a) \ll p_{\mathcal{M}'}(\cdot|s, a)$ and $q_\mathcal{M}(\cdot|s, a) \ll q_{\mathcal{M}'}(\cdot|s, a)$. In that case, for any state (action pair) $(s, a)$, we define the KL divergence of the distributions of the one-step observations under $\mathcal{M}$ and $\mathcal{M}'$ when starting at $(s, a)$ as $\mathrm{KL}_{\mathcal{M}|\mathcal{M}'}(s, a) \triangleq KL(p_\mathcal{M}(\cdot|s, a)\|p_{\mathcal{M}'}(\cdot|s, a)) + KL(q_\mathcal{M}(\cdot|s, a)\|q_{\mathcal{M}'}(\cdot|s, a))$.

## 3 Sample complexity lower bound

In this section, we first derive a lower bound on the expected sample complexity satisfied by any $\delta$-PC algorithm. The lower bound is obtained as the solution of a non-convex optimization problem, as in the case where the learner has access to a generative model [MP21]. The problem has however additional constraints, referred to as the *navigation* constraints, due the fact that the learner has access to a single system trajectory.

The expected sample complexity of an algorithm $\mathbb{A}$ is $\mathbb{E}_{\mathcal{M}, \mathbb{A}}[\tau_\delta] = \sum_{s,a} \mathbb{E}_{\mathcal{M}, \mathbb{A}}[N_{sa}(\tau_\delta)]$. Lower bounds on the sample complexity are derived by identifying constraints that the various $N_{sa}(\tau_\delta)$'s need to satisfy so as to get a $\delta$-PC algorithm. We distinguish here two kinds of constraints:

**Information constraints.** These are constraints on $\mathbb{E}_{\mathcal{M}, \mathbb{A}}[N_{sa}(\tau_\delta)]$, so that the algorithm can learn the optimal policy with probability at least $1 - \delta$. They are the same as those derived [MP21] when the learner has access to a generative model and are recalled in Lemma 9 in Appendix C.

**Navigation constraints.** Observations come from a single (but controlled) system trajectory which imposes additional constraints on the $N_{sa}(\tau_\delta)$'s. These are derived by just writing the Chapman-Kolmogorov equations of the controlled Markov chain (refer to Appendix C for a proof).

**Lemma 1.** *For any algorithm $\mathbb{A}$, and for any state $s$, we have:*

$$\left| \mathbb{E}_{\mathcal{M}, \mathbb{A}}[N_s(\tau_\delta)] - \sum_{s', a'} p_\mathcal{M}(s|s', a') \mathbb{E}_{\mathcal{M}, \mathbb{A}}[N_{s'a'}(\tau_\delta)] \right| \leq 1. \tag{1}$$

Putting all constraints together, we obtain the following sample complexity lower bound.

---

[4]Unless we modify the objective to finding the optimal policy in this subchain.

**Proposition 2.** *Define the set of navigation-constrained allocation vectors:*

$$\Omega(\mathcal{M}) = \Big\{ \omega \in \Sigma : \ \forall s \in \mathcal{S}, \ \omega_s = \sum_{s',a'} p_{\mathcal{M}}(s|s',a')\omega_{s'a'} \Big\}.$$

*Further define* $\mathrm{Alt}(\mathcal{M}) = \{\mathcal{M}' \ MDP : \mathcal{M} \ll \mathcal{M}', \Pi^\star(\mathcal{M}) \cap \Pi^\star(\mathcal{M}') = \emptyset\}$ *the set of alternative instances. Then the expected sample complexity* $\mathbb{E}_{\mathcal{M},\mathbb{A}}[\tau_\delta]$ *of any $\delta$-PC algorithm $\mathbb{A}$ satisfies:*

$$\liminf_{\delta \to 0} \frac{\mathbb{E}_{\mathcal{M},\mathbb{A}}[\tau]}{\log(1/\delta)} \geq T_o(\mathcal{M}), \quad where \quad T_o(\mathcal{M})^{-1} = \sup_{\omega \in \Omega(\mathcal{M})} T(\mathcal{M}, \omega)^{-1} \qquad (2)$$

*and*

$$T(\mathcal{M}, \omega)^{-1} = \inf_{\mathcal{M}' \in \mathrm{Alt}(\mathcal{M})} \sum_{s,a} \omega_{sa} KL_{\mathcal{M}|\mathcal{M}'}(s,a). \qquad (3)$$

**Remark 1.** *A common interpretation of change-of-measure lower bounds like the one above is the following: the optimization problem in the definition of $T_o(\mathcal{M})$ can be seen as the value of a two-player zero-sum game between an algorithm which samples each state-action $(s,a)$ proportionally to $\omega_{sa}$ and an adversary who chooses an alternative instance $\mathcal{M}'$ that is difficult to distinguish from $\mathcal{M}$ under the algorithm's sampling strategy. This suggests that an optimal algorithm should play the optimal allocation that solves the optimization problem (2) and, as a consequence, rules out all alternative instances as fast as possible.*

**Remark 2.** *Note that compared to the lower bound of the generative setting in [MP21], the only change is in the set of sampling strategies that the algorithm can play, which is no longer equal to the entire simplex.*

## 3.1 Proxy for the optimal allocation and the characteristic time

As shown in [MP21], even without accounting for the navigation constraints, computing the characteristic time $T_o(\mathcal{M})$ and in particular, the optimal allocation leading to it is not easy. Indeed, the sub-problem corresponding to computing $T(\mathcal{M}, \omega)^{-1}$ is non-convex. This makes it difficult to design an algorithm that targets the optimal weights $\arg\max_{\omega \in \Omega(\mathcal{M})} T(\mathcal{M}, \omega)^{-1}$. Instead, we use a tractable upper bound of $T(\mathcal{M}, \omega)$ from [MP21]:

**Lemma 3.** *(Theorem 1, [MP21]) For all vectors $\omega \in \Sigma$, $T(\mathcal{M}, \omega) \leq U(\mathcal{M}, \omega)$, where*[5]

$$U(\mathcal{M}, \omega) = \max_{(s,a):a \neq \pi^\star(s)} \frac{H_{sa}}{\omega_{sa}} + \frac{H^\star}{S \min_s \omega_{s,\pi^\star(s)}}, \qquad (4)$$

*and*
$$\begin{cases} H_{sa} = \dfrac{2}{\Delta_{sa}^2} + \max\left( \dfrac{16 \mathrm{Var}_{p(s,a)}[V_{\mathcal{M}}^\star]}{\Delta_{sa}^2}, \dfrac{6 \, \mathrm{sp}(V_{\mathcal{M}}^\star)^{4/3}}{\Delta_{sa}^{4/3}} \right), \\ H^\star = \dfrac{2S}{[\Delta_{\min}(1-\gamma)]^2} + \mathcal{O}\left( \dfrac{S}{\Delta_{\min}^2(1-\gamma)^3} \right). \end{cases} \qquad (5)$$

Using $U(\mathcal{M}, \omega)$, we obtain the following upper bound on the characteristic time (2):

$$T_o(\mathcal{M}) \leq U_o(\mathcal{M}) \triangleq \inf_{\omega \in \Omega(\mathcal{M})} U(\mathcal{M}, \omega). \qquad (6)$$

The advantages of the above upper bound $U_o(\mathcal{M})$ are that: (i) it is a problem-specific quantity as it depends on the gaps and variances of the value function in $\mathcal{M}$; (ii) the corresponding allocation (that solves (6)) can be easily computed, and hence targeted. Indeed, the optimization problem in (6) has convex objective and constraints. Therefore, we can use the projected subgradient-descent algorithm to compute

$$\boldsymbol{\omega}^\star(\mathcal{M}) \triangleq \arg\min_{\omega \in \Omega(\mathcal{M})} U(\mathcal{M}, \omega) = \arg\max_{\omega \in \Omega(\mathcal{M})} U(\mathcal{M}, \omega)^{-1}, \qquad (7)$$

which will be used in our algorithm as a proxy[6] for $\arg\max_{\omega \in \Omega(\mathcal{M})} T(\mathcal{M}, \omega)^{-1}$.

---

[5]The exact definition of $H^\star$ is given in Appendix C.

[6]Note that if we have access to an optimization oracle that, given a MDP $\mathcal{M}$, returns the optimal allocation solution to (2), then we can replace $\boldsymbol{\omega}^\star(\mathcal{M})$ by this optimal allocation. Our algorithm will then be asymptotically optimal up to a factor of 2.

# 4 Algorithm

We propose MDP-NaS (MDP Navigate-and-Stop), a model-based algorithm that is inspired by the lower bound. The lower bound suggests that to identify the best policy in a sample-efficient manner, the algorithm must collect samples from state-action pair $(s, a)$ proportionally to $\omega_{sa}^\star(\mathcal{M})$. We propose two sampling rules which ensure that the former statement holds in the long term (see Section 4.1 and Theorem 7 for a rigorous formulation). Our sampling rules are combined with a Generalized Likelihood Ratio (GLR) test (or rather a proxy of the GLR test, see Section 4.2 for details), that stops as soon as we are confident that $\widehat{\pi}_t^\star = \pi^\star$ with probability at least $1 - \delta$. The pseudo-code for MDP-NaS is given in Algorithm 1.

---

**Algorithm 1:** MDP Navigate and Stop (MDP-NaS)

**Input:** Confidence level $\delta$, ERGODIC boolean variable, communication parameter $m$ or an upper bound.

1 **if** *ERGODIC* **then**
2     Set $(\varepsilon_t)_{t \geq 1} = (1/\sqrt{t})_{t \geq 1}$.
3 **else**
4     Set $(\varepsilon_t)_{t \geq 1} = (t^{-\frac{1}{m+1}})_{t \geq 1}$.
5 **end**
6 Set $t \leftarrow 0$ and $N_{sa}(t) \leftarrow 0$, for all (s,a).
7 Initialize empirical estimate $\widehat{\mathcal{M}}_0$ by drawing an arbitrary MDP from $\mathbb{M}$.
8 Compute $\widehat{\pi}_t^\star \leftarrow$ POLICY-ITERATION$(\widehat{\mathcal{M}}_t)$.
9 **while** *Stopping condition (13) is not satisfied* **do**
10     Compute $\omega^\star(\widehat{\mathcal{M}}_t)$ according to (7) and $\pi^o(\widehat{\mathcal{M}}_t)$ according to (8).
11     **if** $t = 0$ **then**
12        Play $a_0 \sim \text{Unif}([|1, A|])$.
13     **else**
14        Play $a_t \sim \pi_t(.|s_t)$, where $\pi_t$ is determined by either (9) or (10).
15     **end**
16     Observe $(R_t, s_{t+1}) \sim q(.|s_t, a_t) \otimes p(.|s_t, a_t)$.
17     $t \leftarrow t + 1$.
18     Update $\widehat{\mathcal{M}}_t$ and $(N_{sa}(t))_{s,a}$.
19     Compute $\widehat{\pi}_t^\star \leftarrow$ POLICY-ITERATION$(\widehat{\mathcal{M}}_t)$.
20 **end**

**Output:** Empirical optimal policy $\widehat{\pi}_\tau^\star$.

---

## 4.1 Sampling rule

We introduce a few definitions to simplify the presentation. Any stationary policy $\pi$ induces a Markov chain on $\mathcal{Z}$ whose transition kernel is defined by $P_\pi((s, a), (s', a')) \triangleq P(s'|s, a)\pi(a'|s')$. With some abuse of notation, we will use $P_\pi$ to refer to both the Markov chain and its kernel. We denote by $\pi_u$ the uniform random policy, i.e., $\pi_u(a|s) = 1/A$ for all pairs $(s, a)$. Finally, we define the vector of visit-frequencies $\boldsymbol{N}(t)/t \triangleq (N_{sa}(t)/t)_{(s,a) \in \mathcal{Z}}$.

In contrast with pure exploration in Multi-Armed Bandits and MDPs with a generative model where any allocation vector in the simplex is achievable, here the agent can only choose a *sequence of actions* and follow the resulting trajectory. Therefore, one might ask if the oracle allocation can be achieved by following a simple policy. A natural candidate is the *oracle policy* defined by

$$\forall (s, a) \in \mathcal{Z}, \quad \pi^o(a|s) \triangleq \frac{\omega_{sa}^\star}{\omega_s^\star} = \frac{\omega_{sa}^\star}{\sum_{b \in \mathcal{A}} \omega_{sb}^\star}. \tag{8}$$

It is immediate to check that $\omega^\star$ is the stationary distribution of $P_{\pi^o}$. Denote by $\pi^o(\mathcal{M})$ the above oracle policy defined through $\omega^\star(\mathcal{M})$. Policy $\pi^o(\mathcal{M})$ is the target that we would like to play, but since the rewards and dynamics of $\mathcal{M}$ are unknown, the actions must be chosen so as to estimate consistently the oracle policy while at the same time ensuring that $\boldsymbol{N}(t)/t$ converges to $\omega^\star(\mathcal{M})$. The following two sampling rules satisfy these requirements.

**D-Navigation rule:** At time step $t$, the learner plays the policy

$$\pi_t = \varepsilon_t \pi_u + (1 - \varepsilon_t)\pi^o(\widehat{\mathcal{M}}_t), \tag{9}$$

**C-Navigation rule:** At time step $t$, the learner plays

$$\pi_t = \varepsilon_t \pi_u + (1 - \varepsilon_t)\sum_{j=1}^{t} \pi^o(\widehat{\mathcal{M}}_j)/t, \tag{10}$$

where $\varepsilon_t$ is a decreasing exploration rate to be tuned later. In the second case, the agent navigates using a Cesàro-mean of oracle policies instead of the current estimate of the oracle policy, which makes the navigation more stable.

### 4.1.1 Tuning the forced exploration parameter

The mixture with the uniform policy[7] helps the agent to explore all state-action pairs often enough in the initial phase. This is particularly necessary in the case where the empirical weight $\omega^\star_{sa}(\widehat{\mathcal{M}}_t)$ of some state-action pair $(s, a)$ is under-estimated, which may cause that this pair is left aside and hence the estimation never corrected. The next result in this section gives a tuning of the rate $\varepsilon_t$ that ensures a sufficient forced exploration:

**Lemma 4.** *Let $m$ be the maximum length of the shortest paths (in terms of number of transitions) between pairs of states in $\mathcal{M}$: $m \triangleq \max_{(s,s')\in\mathcal{S}^2} \min\{n \geq 1 : \exists \pi : \mathcal{S} \to \mathcal{A}, P^n_\pi(s, s') > 0\}$. Then C-Navigation or D-Navigation with any decreasing sequence $(\varepsilon_t)_{t\geq 1}$ such that $\forall t \geq 1$, $\varepsilon_t \geq t^{-\frac{1}{m+1}}$ satisfies: $\mathbb{P}_{\mathcal{M},\mathbb{A}}\big(\forall(s, a) \in \mathcal{Z}, \lim_{t\to\infty} N_{sa}(t) = \infty\big) = 1$.*

**Remark 3.** *When the parameter $m$ is unknown to the learner, one can replace it by its worst case value $m_{\max} = S - 1$. However, when prior knowledge is available, using a faster-decreasing sequence $\varepsilon_t = t^{-\frac{1}{m+1}}$ instead of $t^{-\frac{1}{S}}$ can be useful to accelerate convergence, especially when the states of $\mathcal{M}$ are densely connected ($m \ll S - 1$).*

**Minimal exploration rate: Communicating MDPs.** The forced exploration rate $t^{-\frac{1}{S}}$ vanishes quite slowly. One may wonder if this rate is necessary to guarantee sufficient exploration in communicating MDPs: the answer is yes in the worst case, as the following example shows. Consider a variant of the classical RiverSwim MDP with state (resp. action) space $\mathcal{S} = [[1, S]]$, (resp. $\mathcal{A} = \{\text{LEFT1}, \text{LEFT2}, \text{RIGHT}\}$). LEFT1 and LEFT2 are equivalent actions inducing the same rewards and transitions. After playing RIGHT the agent makes a transition of one step to the right, while playing LEFT1 (or LEFT2) moves the agent all the way back to state 1. The rewards are null everywhere but in states $\{1, S\}$ where $R(1, \text{LEFT1}), R(1, \text{LEFT2})$ are equal to 0.01 almost surely and $R(S, \text{RIGHT})$ is Bernoulli with mean 0.02.

When $\gamma$ is close to 1, the optimal policy consists in always playing RIGHT so as to reach state $S$ and stay there indefinitely. Now suppose that the agent starts at $s = 1$ and due to the small probability of observing the large reward in state $S$, she underestimates the value of this state in the first rounds and focuses on distinguishing the best action to play in state 1 among $\{\text{LEFT1}, \text{LEFT2}\}$. Under this scenario she ends up playing a sequence of policies that scales like $\pi_t = 2\varepsilon_t\pi_u + (1 - 2\varepsilon_t)(\text{LEFT1} + \text{LEFT2})/2$[8]. This induces the non-homogeneous Markov Chain depicted in Figure 1. For the exploration rate $\varepsilon_t = t^{-\alpha}$, we show that if the agent uses any $\alpha > \frac{1}{S-1}$, with non-zero probability she will visit state $S$ only a finite number of times. Therefore she will fail to identify the optimal policy for small values of the confidence level $\delta$. The proof is deferred to Appendix D.

**Minimal exploration rate: Ergodic MDPs.** For such MDPs, we can select $\varepsilon_t = 1/t^\alpha$ where $\alpha < 1$ without compromising the conclusion of Lemma 4. The proof is deferred to Appendix D.

### 4.2 Stopping rule

To implement a GLR test, we define $\ell_{\mathcal{M}'}(t)$, the likelihood of the observations under some MDP $\mathcal{M}' \ll \mathcal{M}$: $\ell_{\mathcal{M}'}(t) = \prod_{k=0}^{t-1} p_{\mathcal{M}'}(s_{k+1}|s_k, a_k)q_{\mathcal{M}'}(R_k|s_k, a_k)$, where at step $k$ the algorithm is in

---

[7]Our results still hold if $\pi_u$ is replaced by any other policy $\pi$ which has an ergodic kernel. In practice, this can be helpful especially when we have prior knowledge of a fast-mixing policy $\pi$.

[8]Modulo a re-scaling of $\varepsilon_t$ by a constant factor of 1/2.

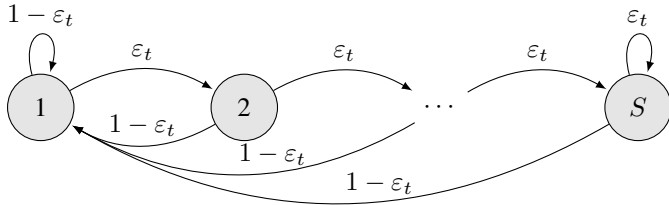

Figure 1: Non-homogeneous Markov Chain. An exploration rate of at least $t^{-\frac{1}{S-1}}$ is needed.

state $s_k$, plays action $a_k$ and observes the reward $R_k$ and $s_{k+1}$ (the next state). Performing a GLR test in step $t$ consists in computing the optimal policy $\widehat{\pi}_t^\star$ for the estimated MDP $\widehat{\mathcal{M}}_t$ and in comparing the likelihood of observations under the most likely model where $\widehat{\pi}_t^\star$ is optimal to the likelihood under the most likely model where $\widehat{\pi}_t^\star$ is sub-optimal. Following the standardized form of the GLR for multi-armed bandits in [DKM19] we write:

$$
G_{\widehat{\pi}_t^\star}(t) = \log \frac{\sup\limits_{\mathcal{M}':\widehat{\pi}_t^\star \in \Pi^\star(\mathcal{M}')} \ell_{\mathcal{M}'}(t)}{\sup\limits_{\mathcal{M}':\widehat{\pi}_t^\star \notin \Pi^\star(\mathcal{M}')} \ell_{\mathcal{M}'}(t)} = \log \frac{\ell_{\widehat{\mathcal{M}}_t}(t)}{\sup\limits_{\mathcal{M}' \in \mathrm{Alt}(\widehat{\mathcal{M}}_t)} \ell_{\mathcal{M}'}(t)} = \inf\limits_{\mathcal{M}' \in \mathrm{Alt}(\widehat{\mathcal{M}}_t)} \log \frac{\ell_{\widehat{\mathcal{M}}_t}(t)}{\ell_{\mathcal{M}'}(t)}
$$

$$
= \inf\limits_{\mathcal{M}' \in \mathrm{Alt}(\widehat{\mathcal{M}}_t)} \sum\limits_{(s,a) \in \mathcal{Z}} N_{sa}(t)\big[\, \mathrm{KL}(\widehat{q}_{s,a}(t),\, q_{\mathcal{M}'}(s,a)) + \mathrm{KL}(\widehat{p}_{s,a}(t),\, p_{\mathcal{M}'}(s,a)) \big] \quad (11)
$$

$$
= t\, T\Big(\widehat{\mathcal{M}}_t,\, \boldsymbol{N}(t)/t\Big)^{-1}. \tag{12}
$$

The hypothesis $(\widehat{\pi}_t^\star \neq \pi^\star)$ is then rejected as soon as the ratio of likelihoods becomes greater than the threshold $\beta(t,\delta)$, properly tuned to ensure that the algorithm is $\delta$-PC. Note that the likelihood ratio $G_{\widehat{\pi}_t^\star}(t)$ itself can be difficult to compute for MDPs, since it is equivalent to solving (3), we lower bound it using (12) and Lemma 3. This leads to the following proposition:

**Proposition 5.** *Define the random thresholds for the transitions and rewards respectively:*

$$
\beta_p(t,\delta) \triangleq \log(1/\delta) + (S-1)\sum\limits_{(s,a)} \log\left(e\big[1 + N_{sa}(t)/(S-1)\big]\right)
$$

$$
\beta_r(t,\delta) \triangleq SA\, \varphi\big(\log(1/\delta)/SA\big) + 3\sum\limits_{s,a} \log\big[1 + \log(N_{sa}(t))\big]
$$

*where $\varphi(x) \underset{\infty}{\sim} x + \log(x)$ is defined in the appendix. Then the stopping rule:*

$$
\tau_\delta \triangleq \inf\left\{t \geq 1 :\; t\, U\big(\widehat{\mathcal{M}}_t,\, \boldsymbol{N}(t)/t\big)^{-1} \geq \beta_r(t,\delta/2) + \beta_p(t,\delta/2)\right\} \tag{13}
$$

*is $\delta$-PC, i.e., $\mathbb{P}(\tau_\delta < \infty,\, \widehat{\pi}_{\tau_\delta}^\star \neq \pi^\star) \leq \delta$.*

## 5 Main results and sample complexity analysis

First we state our main results, which take the form of asymptotic upper bounds on the sample complexity of MDP-NaS, under a slightly more conservative exploration rate than in Lemma 4. Then we present the most important ingredients in their proof. The complete proof is provided in Appendix F.

**Theorem 6.** *Using the C-Navigation sampling rule with $\varepsilon_t = t^{-\frac{1}{2(m+1)}}$ and the stopping rule (13):*
*(i) MDP-NaS stops almost surely and its stopping time satisfies*
$\mathbb{P}\big(\limsup_{\delta \to 0} \frac{\tau_\delta}{\log(1/\delta)} \leq 2U_o(\mathcal{M})\big) = 1$, *where $U_o(\mathcal{M})$ was defined in (6);*
*(ii) the stopping time of MDP-NaS has a finite expectation for all $\delta \in (0,1)$, and*
$\limsup_{\delta \to 0} \frac{\mathbb{E}[\tau_\delta]}{\log(1/\delta)} \leq 2U_o(\mathcal{M})$.

## 5.1 Proof sketch

**Concentration of empirical MDPs:** The starting point of our proof is a concentration event of the empirical estimates $\widehat{\mathcal{M}}_t$ around $\mathcal{M}$. For $\xi > 0$ and $T \geq 1$, we define $\mathcal{C}_T^1(\xi) = \bigcap_{t=T^{1/4}}^T \left( \left\| \widehat{\mathcal{M}}_t - \mathcal{M} \right\| \leq \xi, \left\| \pi^o(\widehat{\mathcal{M}}_t) - \pi^o(\mathcal{M}) \right\|_\infty \leq \xi \right)$, where $\|.\|$ is a semi-norm on MDPs. Then we prove in Lemma 18 that $\mathcal{C}_T^1(\xi)$ holds with high probability in the sense that for all $T \geq 1$:

$$\mathbb{P}\left( \mathcal{C}_T^1(\xi) \right) \geq 1 - \mathcal{O}\left( 1/T^2 \right). \tag{14}$$

For this purpose we derive, for all pairs $(s,a)$, a lower bound on $N_{sa}(t)$ stating that[9]: $\mathbb{P}\left( \forall (s,a) \in \mathcal{Z}, \forall t \geq 1, N_{sa}(t) \geq (t/\lambda_\alpha)^{1/4} - 1 \right) \geq 1 - \alpha$ where $\lambda_\alpha \propto \log^2(1 + SA/\alpha)$ is a parameter that depends on the mixing properties of $\mathcal{M}$ under the uniform policy. These lower bounds on $N_{sa}(t)$ w.h.p. contrast with their deterministic equivalent obtained for C-tracking in [GK16].

**Concentration of visit-frequencies:** Before we proceed, we make the following assumption.

**Assumption 2.** $P_{\pi_u}$ *is aperiodic*[10].

Under Assumptions 1 and 2 the kernel $P_{\pi_u}$, and consequently also $P_{\pi^o}$, becomes ergodic. Hence the iterates of $P_{\pi^o}$ converge to $\omega^\star(\mathcal{M})$ at a geometric speed. Also note that the Markov chains induced by playing either of our sampling rules are non-homogeneous, with a sequence of kernels $(P_{\pi_t})_{t \geq 1}$ that is history-dependent. To tackle this difficulty, we adapt a powerful ergodic theorem (see Proposition 12 in the appendix) from [FMP11] originally derived for adaptive Markov Chain Monte-Carlo (MCMC) algorithms to get the following result.

**Theorem 7.** *Using the C-Navigation or D-Navigation we have:* $\lim_{t \to \infty} \mathbf{N}(t)/t = \omega^\star(\mathcal{M})$ *almost surely.*

Lemma 4 and Theorem 7 combined prove Theorem 6 *(i)* in a straightforward fashion. The proof of Theorem 6 *(ii)* is more involved. Again, we adapt the proof method from [FMP11] to derive a finite-time version of Proposition 12 which results into the following proposition.

**Proposition 8.** *Under C-Navigation, for all $\xi > 0$, there exists a time $T_\xi$ such that for all $T \geq T_\xi$, all $t \geq T^{3/4}$ and all functions $f : \mathcal{Z} \to \mathbb{R}^+$, we have:*

$$\mathbb{P}\left( \left| \frac{\sum_{k=1}^t f(s_k, a_k)}{t} - \mathbb{E}_{(s,a) \sim \omega^\star}[f(s,a)] \right| \geq K_\xi \|f\|_\infty \xi \Big| \mathcal{C}_T^1(\xi) \right) \leq 2 \exp\left( -t\xi^2 \right).$$

*where $\xi \mapsto K_\xi$ is a mapping with values in $(1, \infty)$ such that $\limsup_{\xi \to 0} K_\xi < \infty$.*

Now define $\mathcal{C}_T^2(\xi) = \bigcap_{t=T^{3/4}}^T \left( \left| \mathbf{N}(t)/t - \omega^\star(\mathcal{M}) \right| \leq K_\xi \xi \right)$. Then Proposition 8 and Eq. (14) combined imply that for $T$ large enough, the event $\mathcal{C}_T^1(\xi) \cap \mathcal{C}_T^2(\xi)$ holds w.h.p. so that the expected stopping time is finite on the complementary event: $\mathbb{E}[\tau_\delta \mathbb{1}\{\overline{\mathcal{C}_T^1(\xi)} \cup \overline{\mathcal{C}_T^2(\xi)}\}] < \infty$. Now given the asymptotic shape of the thresholds: $\beta_r(t, \delta/2) + \beta_p(t, \delta/2) \underset{\delta \to 0}{\sim} 2\log(1/\delta)$, we may informally write:

$$\mathbb{E}\left[ \tau_\delta \mathbb{1}\{\mathcal{C}_T^1(\xi) \cap \mathcal{C}_T^2(\xi)\} \right] \underset{\delta \to 0}{\preceq} 2\log(1/\delta) \sup_{(\mathcal{M}', \omega') \in B_\xi} U_o(\mathcal{M}', \omega'),$$

where $B_\xi = \{(\mathcal{M}', \omega') : \|\mathcal{M}' - \mathcal{M}\| \leq \xi, \|\omega' - \omega^\star(\mathcal{M})\|_\infty \leq K_\xi \xi\}$. Taking the limits when $\delta$ and $\xi$ go to zero respectively concludes the proof.

---

[9]Using the method in our proof one can derive a lower bound of the type $t^\zeta$ for any $\zeta < 1/2$ and any exploration rate $\varepsilon_t = t^{-\theta}$ with $\theta < 1/(m+1)$. We chose $\theta = \frac{1}{2(m+1)}$ and $\zeta = 1/4$ because they enable us to have an explicit formula for $\lambda_\alpha$. See Lemma 11 in the appendix.

[10]This assumption is mild as it is enough to have only one state $\tilde{s}$ and one action $\tilde{a}$ such that $P_{\mathcal{M}}(\tilde{s}|\tilde{s}, \tilde{a}) > 0$ for it to be satisfied. Furthermore, Assumptions 1 and 2 combined are still less restrictive than the usual *ergodicity* assumption, which requires that the Markov chains of *all* policies are ergodic.

# 6    Comparison with MOCA

Recall from Lemma 3 and Theorem 6 that our sample complexity bound writes as:

$$\mathcal{O}\left( \inf_{\omega \in \Omega(\mathcal{M})} \max_{(s,a):a \neq \pi^\star(s)} \frac{1 + \text{Var}_{p(s,a)}[V_{\mathcal{M}}^\star]}{\omega_{sa}\Delta_{sa}^2} + \frac{1}{\min\limits_s \omega_{s,\pi^\star(s)}\Delta_{\min}^2 (1-\gamma)^3} \right) \log(1/\delta).$$

Hence, in the asymptotic regime $\delta \to 0$, MDP-NaS finds the optimal way to balance exploration between state-action pairs proportionally to their *hardness*: $(1 + \text{Var}_{p(s,a)}[V_{\mathcal{M}}^\star])/\Delta_{sa}^2$ for sub-optimal pairs (resp. $1/\Delta_{\min}^2(1-\gamma)^3$ for optimal pairs).

After a preprint of this work was published, [WSJ21] proposed MOCA, an algorithm for BPI in the episodic setting. MOCA has the advantage of treating the more general case of $\varepsilon$-optimal policy identification with finite-time guarantees on its sample complexity. The two papers have different and complementary objectives but one can compare with their bound for exact policy identification, i.e when $\varepsilon < \varepsilon^{\star}$[11], in the asymptotic regime $\delta \to 0$. In this case, by carefully inspecting the proofs of [WSJ21], we see that MOCA's sample complexity writes as[12]:

$$\mathcal{O}\left( \sum_{h=1}^{H} \inf_{\omega \in \Omega(\mathcal{M})} \max_{s,a \neq \pi^\star(s)} \frac{H^2}{\omega_{sa}(h)\Delta_{sa}(h)^2} \right) \log(1/\delta) + \frac{\text{polylog}(S, A, H, \log(\varepsilon^*)) \log^3(1/\delta)}{\varepsilon^*}$$

where $H$ is the horizon and $\Delta_{sa}(h)$ is the sub-optimality gap of $(s,a)$ at time step $h$. We make the following remarks about the bounds above:

1. MOCA only pays the cost of worst-case visitation probability multiplied by the gap of the corresponding state $\min\limits_{s,a \neq \pi^\star(s)} \omega_{sa}\Delta_{sa}^2$. Instead, MDP-NaS pays a double worst-case cost of the smallest visitation probability multiplied by the minimum gap $\min_s \omega_{s,\pi^\star(s)}\Delta_{\min}^2$. As pointed out by [WSJ21] the former scaling is better, especially when the state where the minimum gap is achieved is different from the one that is hardest to reach. This is however an artefact of the upper bound in Lemma 3 that MDP-NaS uses as a proxy for the characteristic time. Using a more refined bound, or an optimization oracle that solves the best-response problem, one can remove this double worst-case dependency.

2. The sample complexity of MOCA *divided by* $\log(1/\delta)$ *still explodes* when $\delta$ goes to zero, contrary to MDP-NaS's bound.

3. The sample complexity of MDP-NaS is variance-sensitive for sub-optimal state-action pairs, while MOCA's bound depends on a worst-case factor of $H^2$. Indeed as the rewards are in $[0,1]$, we always have $V_{\mathcal{M}}^\star \leq H$ and $\text{Var}_{p(s,a)}[V_{\mathcal{M}}^\star] \leq H^2$ in the episodic setting.

We conclude this section by noting that MDP-NaS has a simple design and can easily be implemented.

# 7    Conclusion

To the best of our knowledge, this paper is the first to propose an algorithm with *instance-dependent sample complexity* for Best Policy identification (BPI) in the *full online* setting. Our results are encouraging as they show: 1) How the navigation constraints of online RL impact the difficulty of learning a good policy, compared to the more relaxed sampling schemes of Multi-Armed Bandits and MDPs with a generative model. 2) That, provided access to an optimization oracle that solves the information-theoretical lower bound (resp. some convex relaxation of the lower bound), asymptotic optimal (resp. near-optimal) sample complexity is still possible through adaptive control of the trajectory. This opens up exciting new research questions. First, it is intriguing to understand how the mixing times -and not just the stationary distributions- of Markov chains induced by policies impact the sample complexity of BPI in the moderate confidence regime. A second direction would be to extend our contributions to the problem of finding an $\varepsilon$-optimal policy, which is of more practical interest than identifying the best policy.

---

[11]$\varepsilon^{\star}$ is defined in their work as the threshold value for $\varepsilon$ such that the only $\varepsilon$-optimal policy is the best policy. However, when the objective is to find the optimal policy, it is not clear from their paper how one can determine such a value of $\varepsilon$ without prior knowledge of the ground truth instance.

[12]The $\log^3(1/\delta)$ term comes from the sample complexity of their sub-routine FINDEXPLORABLESETS.

## Broader Impact

This paper tries to advance our theoretical understanding of the sample complexity of finding a good policy in Markov Decision Processes, which can be interesting for both theoreticians and practitioners. However, this paper is not oriented towards a specific application, so a wider broader impact discussion is not applicable.

## Acknowledgments and Disclosure of Funding

The authors would like to thank the anonymous reviewers whose comments and questions helped improve the clarity of this manuscript. A. Proutiere's research is supported by the Wallenberg AI, Autonomous Systems and Software Program (WASP) funded by the Knut and Alice Wallenberg Foundation. A. Garivier and A. Al Marjani acknowledge the support of the Project IDEXLYON of the University of Lyon, in the framework of the Programme Investissements d'Avenir (ANR-16-IDEX-0005), and Chaire SeqALO (ANR-20-CHIA-0020).

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
