# A  Symbols

Table 1: Additional notations used in the appendix

| Symbol | Definition |
|---|---|
| $m$ | Maximum length of shortest paths: $\displaystyle\max_{(s,s')\in\mathcal{S}^2}\min\{n\geq 1:\exists\pi:\mathcal{S}\to\mathcal{A},P_\pi^n(s,s')>0\}$ |
| $P_{\pi_u}$ | Transition kernel of the uniform policy |
| $\omega_u$ | Stationary distribution of $P_{\pi_u}$ |
| $r$ | $\min\{\ell\geq 1:\forall(z,z')\in\mathcal{Z}^2,\ P_{\pi_u}^\ell(z,z')>0\}$ |
| $\sigma_u$ | $\displaystyle\min_{z,z'\in\mathcal{Z}}\frac{P_{\pi_u}^r(z,z')}{\omega_u(z')}$ |
| $\eta_1$ | $\min\left\{P_{\pi_u}(z,z')\,\big|\,(z,z')\in\mathcal{Z}^2,P_{\pi_u}(z,z')>0\right\}$ |
| $\eta_2$ | $\min\left\{P_{\pi_u}^n(z,z')\,\big|\,(z,z')\in\mathcal{Z}^2,n\in[\![1,m+1]\!],P_{\pi_u}^n(z,z')>0\right\}$ |
| $\eta$ | Communication parameter $\eta_1\eta_2$ |
| $\omega^\star$ | oracle weights: $\omega^\star\triangleq\displaystyle\arg\max_{\omega\in\Omega(\mathcal{M})}U(\mathcal{M},\omega)$. |
| $\pi^o$ | oracle policy: $\pi(a|s)\triangleq\dfrac{\omega^\star_{sa}}{\displaystyle\sum_{a'\in\mathcal{A}}\omega^\star_{sa'}}$. |
| $\pi_t^o$ | $\pi^o(\widehat{\mathcal{M}}_t)$ |
| $\overline{\pi_t^o}$ | $\sum_{j=1}^t\pi^o(\widehat{\mathcal{M}}_j)/t$ |
| $Z_{\pi^o}$ | $(I-P_{\pi^o}+\mathbb{1}\omega^{\star\mathsf{T}})^{-1}$ |
| $\kappa_\mathcal{M}$ | Condition number $\|Z_{\pi^o}\|_\infty$ |
| $P_t$ | Kernel of the policy $\pi_t$ |
| $\omega_t$ | Stationary distribution of $P_t$ |
| $C_t,\rho_t$ | Constants such that $\forall n\geq 1,\ \|P_t^n(z_0,.)-\omega_t^\mathsf{T}\|_\infty\leq C_t\rho_t^n$ |
| $L_t$ | $C_t(1-\rho_t)^{-1}$ |

# B  Experiments

In this section, we test our algorithm on two small examples. The first instance is an ergodic MDP, with 5 states, 5 actions per state and a discount factor $\gamma=0.7$. The rewards of each state-action pair come from independent Bernoulli distributions with means sampled from the uniform distribution $\mathcal{U}([0,1])$. The transitions kernels were generated following a Dirichlet distribution $\mathcal{D}(1,\ldots,1)$. The second instance is the classical RiverSwim from [SL08], which is communicating but not ergodic. The instance we used has 5 states and 2 actions: {LEFT, RIGHT}, with deterministic transitions and a discount factor $\gamma=0.95$. Rewards are null everywhere but in states $\{1,5\}$ where they are Bernoulli with respective means $r(1,\text{LEFT})=0.05$ and $r(1,\text{RIGHT})=1$. We fix a confidence level $\delta=0.1$, and for each of these MDPs, we run 30 Monte-Carlo simulations of MDP-NaS with

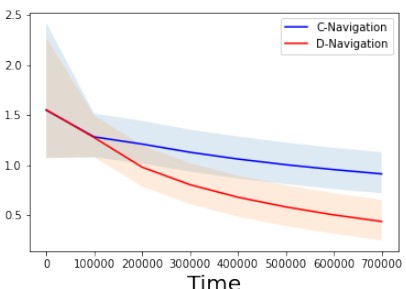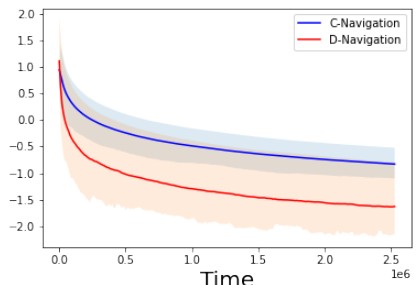

Figure 2: Relative distance in log scale: $\log_{10}\left(\max_{s,a}\frac{|N_{sa}(t)/t-\omega_{sa}^{\star}|}{\omega_{sa}^{\star}}\right)$. Left: Ergodic MDP with $S=A=5$. Right: River Swim with $S=5, A=2$.

either C-Navigation or D-Navigation. Towards computational efficiency, we note that the empirical oracle policy does not change significantly after collecting one sample, therefore we only update it every $10^4$ time steps[13].

First, we seek to check whether the frequencies of state-action pair visits converge to their oracle weights, as stated in Theorem 7. Figure 2 shows the relative distance, in log scale, between the vector of empirical frequencies $\boldsymbol{N}(t)/t$ and the oracle allocation $\boldsymbol{\omega}^{\star}$. The shaded area represents the 10% and 90% quantiles. We see that the relative distance steadily decreases with time, indicating that the visit-frequencies of both D-Navigation and C-Navigation converge to the oracle allocation. We also note that the D-Navigation rule exhibits a faster convergence than the C-Navigation rule. Next we compare our algorithm with Variance-reduced-Q-learning (VRQL) [LWC$^+$20b], a variant of the classical Q-learning with faster convergence rates. VRQL finds an $\varepsilon$-estimate $\widehat{Q}$ of the Q function $Q^{\star}$, by following a fixed sampling rule, referred to as the *behavior policy* $\pi_b$, and updating its estimate of the Q function via Temporal Difference learning. VRQL does not have a stopping rule, but is guaranteed to yield an estimate such that $\left\|\widehat{Q}-Q\right\|_{\infty}\leq\varepsilon$ with probability $1-\delta$ after using $M(N+t_{\text{epoch}})$ samples where

$$M = c_3\log(1/\varepsilon^2(1-\gamma)^2),$$

$$t_{\text{epoch}} = \frac{c_2}{\mu_{\min}}\left(\frac{1}{(1-\gamma)^3}+\frac{t_{mix}}{1-\gamma}\right)\log\left(1/(1-\gamma)^2\varepsilon\right)\log\left(SA/\delta\right),$$

$$N = \frac{c_1}{\mu_{\min}}\left(\frac{1}{(1-\gamma)^3\min(1,\varepsilon^2)}+t_{mix}\right)\log\left(SAt_{\text{epoch}}/\delta\right),$$

where $\mu_{\min}$ (resp. $t_{mix}$) is the minimum state-action occupancy (resp. the mixing time) of $\pi_b$ and $c_1, c_2, c_3$ are some large enough universal constants. We use VRQL with $c_1 = c_2 = c_3 = 10$, $\varepsilon \leq \Delta_{\min}$ (since the goal is to identify the best policy) and use the uniform policy as a sampling rule[14]. We plug this value into the equations above and the compute the sample complexity of VRQL. Table 2 shows a comparison of the sample complexities of MDP-NaS and VRQL. MDP-NaS has much better performance than VRQL.

|  | MDP-NaS | VRQL |
|---|---|---|
| Small Ergodic MDP | $8\times 10^5$ | $2.5\times 10^8$ |
| RIVER-SWIM | $2.6\times 10^6$ | $3.3\times 10^9$ |

Table 2: Average sample complexity of MDP-NaS (D-Navigation) vs deterministic sample complexity of VRQL. $\delta = 0.1$.

---

[13]The period was chosen so as to save computation time, and knowing that for MDPs algorithms usually require $\geq 10^6$ samples to return a reasonably good policy.

[14]In the absence of prior knowledge about the MDP, the uniform policy is a reasonable choice to maximize $\mu_{\min}$.

# C  Sample complexity lower bound

Let $\mathrm{Alt}(\mathcal{M})$ be the set of MDPs such that $\mathcal{M} \ll \mathcal{M}'$ and $\Pi^\star(\mathcal{M}) \cap \Pi^\star(\mathcal{M}') = \emptyset$. The information constraints are obtained by change-of-measure arguments as in the bandit literature [LR85, KCG16]:

**Lemma 9.** *([MP21]) For any $\delta$-PC algorithm $\mathbb{A}$, and for any $\mathcal{M}' \in \mathrm{Alt}(\mathcal{M})$, we have:*

$$\sum_{s,a} \mathbb{E}_{\mathcal{M},\mathbb{A}}[N_{sa}(\tau_\delta)] KL_{\mathcal{M}|\mathcal{M}'}(s,a) \geq \mathrm{kl}(\delta, 1-\delta). \tag{15}$$

## C.1  Navigation constraints: proof of Lemma 1

For all states $s$,

$$N_\tau(s) = \mathbb{1}_{\{S_1=s\}} + \sum_{s',a'} \sum_{u=1}^{N_{\tau-1}(s',a')} \mathbb{1}_{\{W_u=s\}},$$

where $W_u$ denotes the state observed after the $u$-th times $(s',a')$ has been visited. Fix $s', a'$. Introduce $G_t^{s',a'} = \sum_{u=1}^{N_{t-1}(s',a')} \mathbb{1}_{\{W_u=s\}}$. Observe that $\{W_u = s\}$ and $\{N_{t-1}(s',a') > u - 1\}$ are independent. Furthermore, $\mathbb{E}_{\mathcal{M},\mathbb{A}}[\mathbb{1}_{\{W_s=s\}}] = p_{\mathcal{M}}(s|s',a')$. Hence:

$$\mathbb{E}_{\mathcal{M},\mathbb{A}}[G_\tau^{s',a'}] = p_{\mathcal{M}}(s|s',a') \mathbb{E}_{\mathcal{M},\mathbb{A}}[N_{\tau-1}(s',a')].$$

Finally,

$$\mathbb{E}_{\mathcal{M},\mathbb{A}}[N_\tau(s)] = \mathbb{P}_{\mathcal{M}}[\mathbb{1}_{\{S_1=s\}}] + \sum_{s',a'} p_{\mathcal{M}}(s|s',a') \mathbb{E}_{\mathcal{M},\mathbb{A}}[N_{\tau-1}(s',a')]. \tag{16}$$

From the above equality, the lemma is proved by just observing that $\mathbb{P}_{\mathcal{M}}[\mathbb{1}_{\{S_1=s\}}] \leq 1$, $\mathbb{E}_{\mathcal{M},\mathbb{A}}[N_{\tau-1}(s',a')] \leq \mathbb{E}_{\mathcal{M},\mathbb{A}}[N_\tau(s',a')]$ for any $(s',a')$, and $\mathbb{E}_{\mathcal{M},\mathbb{A}}[N_\tau(s)] \leq \mathbb{E}_{\mathcal{M},\mathbb{A}}[N_{\tau-1}(s)] + 1$ for any $s$.

## C.2  Lower bound: proof of Proposition 2

By combining Lemma 9 and Lemma 1 we get the following proposition.

**Proposition 10.** *The expected sample complexity $\mathbb{E}_{\mathcal{M},\mathbb{A}}[\tau_\delta]$ of any $\delta$-PC algorithm $\mathbb{A}$ is larger than the value of the following optimization problem:*

$$\inf_{n \geq 0} \sum_{s,a} n_{sa} \tag{17}$$

$$s.t. \ \forall s, \ \left| \sum_a n_{sa} - \sum_{s',a'} p_{\mathcal{M}}(s|s',a')n_{s'a'} \right| \leq 1,$$

$$\forall \mathcal{M}' \in \mathrm{Alt}(\mathcal{M}), \ \sum_{s,a} n_{sa} KL_{\mathcal{M}|\mathcal{M}'}(s,a) \geq \mathrm{kl}(\delta, 1-\delta).$$

In (17), $n_{sa}$ is interpreted as the expected number of times $(s,a)$ is visited before the stopping time. Note that the above proposition provides a lower bound for any value of $\delta$. We can further simplify this bound when restricting our attention to asymptotic regimes where $\delta$ goes to 0. In that case, the navigation constraints (1) can be replaced by $\sum_a n_{sa} = \sum_{s',a'} p_{\mathcal{M}}(s|s',a')n_{s'a'}$ (by just renormalizing $n$ with $n/\log(1/\delta)$, and letting $\delta \to 0$). For small $\delta$ regimes, we can hence rewrite the lower bound as follows:

$$\liminf_{\delta \to 0} \frac{\mathbb{E}_{\mathcal{M},\mathbb{A}}[\tau]}{\log(1/\delta)} \geq \inf_{n \geq 0} \sum_{s,a} n_{sa}$$

$$s.t. \ \forall s, \ \sum_a n_{sa} = \sum_{s',a'} p_{\mathcal{M}}(s|s',a')n_{s'a'},$$

$$\forall \mathcal{M}' \in \mathrm{Alt}(\mathcal{M}), \ \sum_{s,a} n_{sa} KL_{\mathcal{M}|\mathcal{M}'}(s,a) \geq 1.$$

One can easily conclude by showing that the value of the optimization program above is equal to the one in Eq 2.

## C.3 Full definition of the terms in the upper bound

Let $\mathrm{Var}^\star_{\max}[V^\star_{\mathcal{M}}] = \max_{s \in \mathcal{S}} \mathrm{Var}_{s' \sim p_{\mathcal{M}}(.|s,\pi^\star(s))}[V^\star_{\mathcal{M}}(s')]$ denote the maximum variance of the value function on the trajectory of the optimal policy. In [MP21] are defined the following functionals of $\mathcal{M}$:

$$T_3(\mathcal{M}) \triangleq \frac{2}{\Delta^2_{\min}(1-\gamma)^2},$$

$$T_4(\mathcal{M}) \triangleq \min\left(\frac{27}{\Delta^2_{\min}(1-\gamma)^3}, \max\left(\frac{16\mathrm{Var}^\star_{\max}[V^\star_{\mathcal{M}}]}{\Delta^2_{\min}(1-\gamma)^2}, \frac{6\,\mathrm{sp}(V^\star_{\mathcal{M}})^{4/3}}{\Delta^{4/3}_{\min}(1-\gamma)^{4/3}}\right)\right).$$

Then $H^\star$ is simply defined as $H^\star = S(T_3(\mathcal{M}) + T_4(\mathcal{M}))$. Note that $T_4(\mathcal{M}) = \mathcal{O}\left(\frac{1}{\Delta^2_{\min}(1-\gamma)^3}\right)$.

# D   Sampling rule

Recall that $\mathcal{Z} \triangleq \mathcal{S} \times \mathcal{A}$ denotes the set of state-action pairs. Any policy $\pi$ induces a Markov Chain on $\mathcal{Z}$ whose kernel is defined by:

$$P_\pi((s,a),(s',a')) = P(s'|s,a)\pi(a'|s').$$

**Fact 1:**   Note that if it takes $m$ steps to move between any pair of states $(s,s')$ with non-zero probability, then we can move between any pair of state-actions $((s,a),(s',a'))$ in at most $m+1$ steps by playing the policy:

$$\tilde{\pi}(x) = \begin{cases} a' & \text{if } x = s', \\ \pi_{\hookrightarrow s'}(x) & \text{otherwise} \end{cases}$$

where $\pi_{\hookrightarrow s'}$ is the policy corresponding to shortest path to $s'$. Finally, for the sake of simplicity, we will note $P_t \triangleq P_{\pi_t}$.

## D.1   Almost sure forced exploration: proof of Lemma 4

Consider the event $\mathcal{E} \triangleq \left(\exists z \in \mathcal{Z}, \exists M > 0, \forall t \geq 1, N_z(t) < M\right)$. Observe that $\mathcal{E} = \bigcup_{z \in \mathcal{Z}} \mathcal{E}_z$, where for $z \in \mathcal{Z}$, $\mathcal{E}_z \triangleq \left(\exists M > 0, \forall t \geq 1, N_z(t) < M\right)$. We will prove that $\mathbb{P}(\mathcal{E}_{z'}) = 0$ for all $z'$, which implies the desired result. From Fact 1 above, we have:

$$\forall (z,z') \in \mathcal{Z}^2, \exists r \in [|1, m+1|], \exists \pi \in \Pi, P^r_\pi(z,z') > 0, \tag{18}$$

where $P^r_\pi$ is the $r$-th power of the transition matrix induced by policy $\pi$. Therefore:

$$\eta = \min_{z,z'} \max_{\substack{1 \leq r \leq m+1 \\ \pi \in \Pi}} P^r_\pi(z,z') > 0$$

is well defined. Fix $z \in \mathcal{Z}$ and let $\pi, r_z$ be a policy and an integer satisfying the property (18) above for the pair $(z,z')$. Observe that:

$$P_t \geq \varepsilon_t P_{\pi_u} \geq \frac{\varepsilon_t}{A} P_\pi$$

where the matrix inequality is entry-wise. Now define the stopping times $(\tau_k(z))_{k \geq 1}$ where the agent reaches state-action $z$ for the $k$-th time[15]. Then:

$$\mathbb{P}\big(\mathcal{E}_{z'} \,|\, (\pi_t)_{t \geq 1},\, (\tau_k(z))_{k \geq 1}\big) \leq \mathbb{P}\big(\exists N \geq 1, \forall k \geq N, X_{\tau_k(z)+r_z} \neq z' \,|\, (\pi_t)_{t \geq 1},\, (\tau_k(z))_{k \geq 1}\big)$$

$$\leq \sum_{N=1}^{\infty} \prod_{k=N}^{\infty} \mathbb{P}\big(X_{\tau_k(z)+r_z} \neq z' \,|\, \tau_k(z),\, (\pi_t)_{t \in [|\tau_k(z)+1, \tau_k(z)+r_z|]}\big)$$

$$= \sum_{N=1}^{\infty} \prod_{k=N}^{\infty} \left[1 - \left(\prod_{t=\tau_k(z)+1}^{\tau_k(z)+r_z} P_{\pi_t}\right)(z, z')\right]$$

$$\leq \sum_{N=1}^{\infty} \prod_{k=N}^{\infty} \left[1 - \left(\prod_{t=\tau_k(z)+1}^{\tau_k(z)+r_z} \frac{\varepsilon_t}{A} P_\pi\right)(z, z')\right]$$

$$\leq \sum_{N=1}^{\infty} \prod_{k=N}^{\infty} \left[1 - \frac{\eta}{A^{r_z}} \prod_{t=\tau_k(z)+1}^{\tau_k(z)+r_z} \varepsilon_t\right]$$

$$\leq \sum_{N=1}^{\infty} \prod_{k=N}^{\infty} \left[1 - \frac{\eta}{A^{m+1}} \prod_{t=\tau_k(z)+1}^{\tau_k(z)+m+1} \varepsilon_t\right].$$

The second inequality comes from a union bound and the strong Markov property[16]. The last inequality comes from the fact that $r_z \leq m + 1$ and $\varepsilon_t \leq 1$. Now observe that the inequality above holds for all realizations of the sequences $(\pi_t)_{t \geq 1}$. Therefore, integrating that inequality over all possible sequences of policies yields:

$$\forall z \in \mathcal{Z},\ \mathbb{P}\big(\mathcal{E}_{z'} \,|\, (\tau_k(z))_{k \geq 1}\big) \leq \sum_{N=1}^{\infty} \prod_{k=N}^{\infty} \left[1 - \frac{\eta}{A^{m+1}} \prod_{t=\tau_k(z)+1}^{\tau_k(z)+m+1} \varepsilon_t\right].$$

We can already see that if state-action $z$ is visited "frequently enough" ($\tau_k(z) \sim c.k$ for some constant $c$) then the right-hand side above will be zero. Since we know that a least one state-action $z$ is visited frequently enough, we consider the product over all state-action pairs $z$ of the probabilities above:

$$\prod_{z \in \mathcal{Z}} \mathbb{P}\left(\mathcal{E}_{z'} \,|\, (\tau_k(z))_{k \geq 1}\right) \leq \sum_{(N_1, \ldots, N_{SA}) \in (\mathbb{N}^\star)^{SA}} \prod_{z \in \mathcal{Z}} \prod_{k=N_z}^{\infty} \left[1 - \frac{\eta}{A^{m+1}} \prod_{t=\tau_k(z)+1}^{\tau_k(z)+m+1} \varepsilon_t\right] \quad (19)$$

$$\triangleq \sum_{(N_1, \ldots, N_{SA})} a_{(N_1, \ldots, N_{SA})}.$$

We will now show that $a_{(N_1, \ldots, N_{SA})} = 0$ for all tuples $(N_1, \ldots, N_{SA})$:

$$a_{(N_1, \ldots, N_{SA})} \leq \prod_{z \in \mathcal{Z}} \prod_{k=\max_z N_z}^{\infty} \left[1 - \frac{\eta}{A^{m+1}} \prod_{t=\tau_k(z)+1}^{\tau_k(z)+m+1} \varepsilon_t\right]$$

$$= \prod_{k=\max_z N_z}^{\infty} \prod_{z \in \mathcal{Z}} \left[1 - \frac{\eta}{A^{m+1}} \prod_{t=\tau_k(z)+1}^{\tau_k(z)+m+1} \varepsilon_t\right].$$

Now observe that for all $k \geq 1$ there exists $z_k \in \mathcal{Z}$ such that $\tau_k(z_k) \leq SAk$, i.e., at least one state-action has been visited $k$ times before time step $SAk$. For that particular choice of $z_k$ and since

---

[15]We restrict our attention to departure state-action pairs $z$ that are visited infinitely often. Such pairs always exist, therefore $\tau_k(z)$ is well defined.

[16]This property is sometimes referred to as: Markov Chains start afresh after stopping times.

$(\varepsilon_t)_{t\geq 1}$ is decreasing, we get:

$$a_{(N_1,\ldots,N_{SA})} \leq \prod_{k=\max_z N_z}^{\infty} \left[ 1 - \frac{\eta}{A^{m+1}} \prod_{t=\tau_k(z_k)+1}^{\tau_k(z_k)+m+1} \varepsilon_t \right]$$

$$\leq \prod_{k=\max_z N_z}^{\infty} \left[ 1 - \frac{\eta}{A^{m+1}} \prod_{t=SA.k+1}^{SA.k+m+1} \varepsilon_t \right].$$

For the choice of $\varepsilon_t = t^{-\frac{1}{m+1}}$ the right-hand side above is zero. To sum up, for all realizations of $(\tau_k(z))_{z\in\mathcal{Z}, k\geq 1}$:

$$\prod_{z\in\mathcal{Z}} \mathbb{P}\left( \mathcal{E}_{z'} \,|\, (\tau_k(z))_{k\geq 1} \right) = 0 .$$

Therefore, for all $z'$: $\mathbb{P}(\mathcal{E}_{z'}) = 0$ and consequently: $\mathbb{P}(\mathcal{E}) = 0$.

## D.2 Minimal exploration rate for communicating MDPs

Indeed if the agent visits state $S$ at time $k$, then the last $S-1$ transitions before $k$ must have been to the right, ie $\mathbb{P}(s_k = S) \leq \prod_{j=k-S+1}^{k-1} \varepsilon_j \leq (\varepsilon_{k-S+1})^{S-1}$. Therefore $\mathbb{E}[N_S(t)] \leq \sum_{k=S}^{t}(k-S+1)^{-\alpha(S-1)}$. In particular this implies that for $\alpha > \frac{1}{S-1}$, $\limsup_{t\to\infty} \mathbb{E}[N_S(t)] = M < \infty$. Therefore using the reverse Fatou lemma and Markov's inequality:

$$\mathbb{P}(\forall t \geq 1, N_S(t) \leq 2M) = \mathbb{E}\Big[ \limsup_{t\to\infty} \prod_{k=1}^{t} \mathbb{1}\{N_S(k) \leq 2M\} \Big]$$

$$\geq \limsup_{t\to\infty} \mathbb{E}\Big[ \prod_{k=1}^{t} \mathbb{1}\{N_S(k) \leq 2M\} \Big] = \limsup_{t\to\infty} \mathbb{E}\big[ \mathbb{1}\{N_S(t) \leq 2M\} \big] \geq \frac{1}{2}.$$

## D.3 Minimal exploration rate for ergodic MDPs

This is a consequence of Proposition 2 [BK97], stating that there exist $c_1, c_2, C > 0$ such that for all $s$ and $t$ large enough, $\mathbb{P}_{\mathcal{M},\mathbb{A}}[N_s(t) > c_1 t] \geq 1 - Ce^{-c_2 t}$. A union bound yields: $\mathbb{P}_{\mathcal{M},\mathbb{A}}[\forall s, N_s(t) > c_1 t] \geq 1 - CSe^{-c_2 t}$. To extend this result to the numbers of visits at the various (state, action) pairs, we can derive a lower bound on $N_{sa}(t)$ given that $N_s(t) > c_1 t$ by observing that a worst scenario (by monotonicity of $\varepsilon_s$) occurs when $s$ is visited only in the $c_1 t$ rounds before $t$. We get $\mathbb{E}[N_{sa}(t)|N_s(t) > c_1 t] \geq c_3 t^{(1-\alpha)}$. Remarking that $N_{sa}(t+1) - N_s(t)\varepsilon_t$ is a sub-martingale with bounded increments, standard concentration arguments then imply that $\mathbb{P}_{\mathcal{M},\mathbb{A}}[\forall s, a, N_{sa}(t) > \frac{c_3}{2} t^{(1-\alpha)}] \geq \phi(t)$, where $\phi(t) \to 1$. Next, define the random variable $Z_t = \prod_{s,a} \mathbb{1}\{N_{sa}(t) > \frac{c_3}{2} t^{(1-\alpha)}\}$. Applying the reverse Fatou lemma, we get $1 = \limsup_t \mathbb{E}[Z_t] \leq \mathbb{E}[\limsup_t Z_t]$. From there, we directly deduce (by monotonicity of $t \mapsto N_{sa}(t)$) that a.s. $\lim_{t\to\infty} N_{sa}(t) = \infty$.

## D.4 High probability forced exploration

**Lemma 11.** *Denote by $\tau_k(z)$ the k-th time the agent visits the state-action pair $z$. Under C-Navigation with exploration rate $\varepsilon_t = t^{-\frac{1}{2(m+1)}}$ we have: for all $\alpha \in (0,1)$, there exists a parameter $\eta > 0$ that only depends on $\mathcal{M}$ such that:*

$$\mathbb{P}\left( \forall z \in \mathcal{Z}, \, \forall k \geq 1, \, \tau_k(z) \leq \lambda_\alpha k^4 \right) \geq 1 - \alpha,$$

*where $\lambda_\alpha \triangleq \frac{(m+1)^2}{\eta^2} \log^2(1 + \frac{SA}{\alpha})$.*

**Corollary 1.** *Denote by $N_z(t)$ the number of times the agent visits state-action $z$ up to and including time step $t$. Then under the same notations of the lemma above we have: for all $\alpha \in (0,1)$:*

$$\mathbb{P}\left(\forall z \in \mathcal{Z}, \, \forall t \geq 1, \, N_z(t) \geq \left(\frac{t}{\lambda_\alpha}\right)^{1/4} - 1\right) \geq 1 - \alpha.$$

*Proof.* Let $f$ be some increasing function such that $f(\mathbb{N}) \subset \mathbb{N}$ and $f(0) = 0$ and define the event $\mathcal{E} = \left(\forall z \in \mathcal{Z}, \, \forall k \geq 1, \, \tau_k(z) \leq f(k)\right)$. We will prove the following more general result:

$$\mathbb{P}(\mathcal{E}^c) \leq SA \sum_{k=1}^{\infty} \prod_{j=0}^{\lfloor \frac{f(k)-f(k-1)-1}{m+1} \rfloor - 1} \left[1 - \eta \prod_{l=1}^{m+1} \varepsilon_{f(k-1)+(m+1).j+l}\right], \tag{20}$$

where $\eta$ is a constant depending on the communication properties of $\mathcal{M}$. Then we will tune $f(k)$ and $\varepsilon_t$ so that the right-hand side is less than $\alpha$. First, observe that:

$$\mathcal{E}^c = \bigcup_{z \in \mathcal{Z}} \bigcup_{k=1}^{\infty} \left(\tau_k(z) > f(k) \text{ and } \forall j \leq k-1, \, \tau_j(z) \leq f(j)\right).$$

Using the decomposition above, we upper bound the probability of $\mathcal{E}^c$ by the sum of probabilities for $k \geq 1$ that the $k$-th excursion from and back to $z$ takes too long:

$$\mathbb{P}(\mathcal{E}^c) \leq \sum_{z \in \mathcal{Z}} \left[\mathbb{P}(\tau_1(z) > f(1)) + \sum_{k=2}^{\infty} \mathbb{P}\left(\tau_k(z) > f(k) \text{ and } \forall j \leq k-1, \, \tau_j(z) \leq f(j)\right)\right]$$

$$\leq \sum_{z \in \mathcal{Z}} \left[\mathbb{P}(\tau_1(z) > f(1)) + \sum_{k=2}^{\infty} \mathbb{P}\left(\tau_k(z) > f(k) \text{ and } \tau_{k-1}(z) \leq f(k-1)\right)\right]$$

$$\leq \sum_{z \in \mathcal{Z}} \left[\mathbb{P}(\tau_1(z) > f(1)) + \sum_{k=2}^{\infty} \mathbb{P}\left(\tau_k(z) - \tau_{k-1}(z) > f(k) - f(k-1), \, \tau_{k-1}(z) \leq f(k-1)\right)\right]$$

$$\leq \sum_{z \in \mathcal{Z}} \left[\mathbb{P}(\tau_1(z) > f(1)) + \sum_{k=2}^{\infty} \sum_{n=1}^{f(k-1)} \mathbb{P}\left(\tau_k(z) - \tau_{k-1}(z) > f(k) - f(k-1) \big| \tau_{k-1}(z) = n\right) \mathbb{P}(\tau_0(z) = n)\right]$$

$$= \sum_{z \in \mathcal{Z}} \left[a_1(z) + \sum_{k=2}^{\infty} \sum_{n=1}^{f(k-1)} a_{k,n}(z) \mathbb{P}(\tau_{k-1}(s) = n)\right], \tag{21}$$

where

$$a_1(z) \triangleq \mathbb{P}(\tau_1(z) > f(1)),$$

$$\forall k \geq 2 \, \forall n \in [\![1, f(k-1)]\!], \, a_{k,n}(z) \triangleq \mathbb{P}\left(\tau_k(z) - \tau_{k-1}(z) > f(k) - f(k-1) \big| \tau_{k-1}(z) = n\right).$$

We will now prove an upper bound on $a_{k,n}(z)$ for a fixed $z \in \mathcal{Z}$ and $k \geq 1$.

**1) Upper bounding the probability that an excursion takes too long:** Let us rewrite $P_t$ as

$$P_t = \left(\begin{array}{c|c} Q_t(z) & [P_t(z', z)]_{z' \neq z} \\ \hline [P_t(z, z')]_{z' \neq z}^T & P_t(z, z) \end{array}\right),$$

so that state-action $z$ corresponds to the last row and last column. Further let $p_t(z', \neg z) \triangleq [P_t(z', z")]_{z" \neq z}$ denote the vector of probabilities of transitions at time $t$ from $z'$ to states $z"$ different from $z$. Using a simple recurrence on $N$, one can prove that for all $k, N, n \geq 1$ we have:

$$\mathbb{P}\left(\tau_k(z) - \tau_{k-1}(z) > N \big| \tau_{k-1}(z) = n\right) = p_n(z, \neg z)^T \left(\prod_{j=n+1}^{n+N-1} Q_j(z)\right) \mathbb{1}. \tag{22}$$

Using Lemma 20, there exists $\eta > 0$ (that only depends on $\mathcal{M}$) such that for all $n \geq 1$ and all sequences $(\pi_t)_{t \geq 1}$ such that $\pi_t \geq \varepsilon_t \pi_u$ we have:

$$\left\| \prod_{l=n+1}^{n+m+1} Q_l(z) \right\|_{\infty} \leq 1 - \eta \prod_{l=n+1}^{n+m+1} \varepsilon_l \, . \tag{23}$$

Therefore using (22) for $N = f(k) - f(k-1)$ and breaking the matrix product into smaller product terms of $(m+1)$ matrices, we get for $k \geq 2$:

$$a_{k,n}(z) = \mathbb{P}\left( \tau_k(z) - \tau_{k-1}(z) > f(k) - f(k-1) \Big| \tau_{k-1}(z) = n \right)$$

$$= \mathbb{E}_{(\pi_t)_{t \geq 1}}\left[ \mathbb{P}\left( \tau_k(s) - \tau_{k-1}(s) > f(k) - f(k-1) \Big| \tau_{k-1}(z) = n, (\pi_t)_{t \geq 1} \right) \right]$$

$$= \mathbb{E}_{(\pi_t)_{t \geq 1}}\left[ p_n(z, \neg z)^{\mathsf{T}} \left( \prod_{j=n+1}^{n+f(k)-f(k-1)-1} Q_j(z) \right) \mathbb{1} \right]$$

$$\leq \left\| \prod_{l=n+1}^{n+f(k)-f(k-1)-1} Q_l(z) \right\|_{\infty}$$

$$\leq \left\| \prod_{l=(m+1)\lfloor \frac{f(k)-f(k-1)-1}{m+1} \rfloor +1}^{f(k)-f(k-1)-1} Q_{n+l}(z) \right\|_{\infty} \times \prod_{j=0}^{\lfloor \frac{f(k)-f(k-1)-1}{m+1} \rfloor -1} \left\| \prod_{l=1}^{m+1} Q_{n+(m+1)j+l}(z) \right\|_{\infty}$$

$$\leq \prod_{j=0}^{\lfloor \frac{f(k)-f(k-1)-1}{m+1} \rfloor -1} \left[ 1 - \eta \prod_{l=1}^{m+1} \varepsilon_{n+(m+1)j+l} \right]$$

$$\leq \prod_{j=0}^{\lfloor \frac{f(k)-f(k-1)-1}{m+1} \rfloor -1} \left[ 1 - \eta \prod_{l=1}^{m+1} \varepsilon_{f(k-1)+(m+1)j+l} \right] \triangleq b_k \, , \tag{24}$$

where in the fourth line we used that $\|p_n(z, \neg z)\|_1 \leq 1$. The sixth line uses the fact that the matrices $Q$ are substochastic. The last line is due to the fact that $n \leq f(k-1)$ and $t \mapsto \varepsilon_t$ is decreasing. Similarly, one can prove that:

$$a_1(z) \leq \prod_{j=0}^{\lfloor \frac{f(1)-1}{m+1} \rfloor -1} \left[ 1 - \eta \prod_{l=1}^{m+1} \varepsilon_{(m+1)j+l} \right]$$

$$= \prod_{j=0}^{\lfloor \frac{f(1)-f(0)-1}{m+1} \rfloor -1} \left[ 1 - \eta \prod_{l=1}^{m+1} \varepsilon_{f(0)+(m+1)j+l} \right] \triangleq b_1 \, , \tag{25}$$

where we used the fact that $f(0) = 0$. Now we only have to tune $f(k)$ and $\varepsilon_t$ so that $\sum_{k=1}^{\infty} b_k < \frac{\alpha}{SA}$ and conclude using (21), (24) and (25).

**2) Tuning $f$ and the exploration rate:**  Since the sequence $(\varepsilon_t)_{t \geq 1}$ is decreasing we have:

$$b_k = \prod_{j=0}^{\lfloor \frac{f(k)-f(k-1)-1}{m+1} \rfloor -1} \left[ 1 - \eta \prod_{l=1}^{m+1} \varepsilon_{f(k-1)+(m+1)j+l} \right]$$

$$\leq \prod_{j=0}^{\lfloor \frac{f(k)-f(k-1)-1}{m+1} \rfloor -1} \left[ 1 - \eta \left( \varepsilon_{f(k-1)+(m+1)j+S} \right)^{m+1} \right]$$

$$\leq \left[ 1 - \eta \left( \varepsilon_{f(k)} \right)^{m+1} \right]^{\lfloor \frac{f(k)-f(k-1)-1}{m+1} \rfloor} \, .$$

For $f(k) = \lambda.k^4$ where $\lambda \in \mathbb{N}^\star$ and $\varepsilon_t = t^{-\frac{1}{2(m+1)}}$ we have: $\lfloor \frac{f(k)-f(k-1)-1}{m+1} \rfloor \geq \frac{\lambda k^3}{(m+1)}$ and $\left(\varepsilon_{f(k)}\right)^{m+1} = \frac{1}{\sqrt{\lambda}k^2}$, implying:

$$b_k \leq \left[1 - \frac{\eta}{\sqrt{\lambda}k^2}\right]^{\frac{\lambda k^3}{(m+1)}}$$

$$\leq \exp\left(\frac{-\lambda k^3 \eta}{(m+1)\sqrt{\lambda}k^2}\right) = \exp\left(-\frac{\lambda^{1/2}k\eta}{m+1}\right).$$

Summing the last inequality, along with (21), (24) and (25) we get:

$$\mathbb{P}(\mathcal{E}^c) \leq SA\sum_{k=1}^{\infty} b_k$$

$$\leq SA\sum_{k=1}^{\infty} \exp\left(-\frac{\lambda^{1/2}k\eta}{m+1}\right)$$

$$= \frac{SA\exp\left(-\frac{\lambda^{1/2}\eta}{m+1}\right)}{1 - \exp\left(-\frac{\lambda^{1/2}\eta}{m+1}\right)} \triangleq g(\lambda).$$

For $\lambda_\alpha \triangleq \frac{(m+1)^2}{\eta^2}\log^2(1 + \frac{SA}{\alpha})$, we have $g(\lambda_\alpha) = \alpha$, which gives the desired result.

**Remark 4.** *It is natural that $\lambda$ depends on $\eta$, which expresses how well connected is the MDP under the uniform policy, see proof of Lemma 20.*

$\square$

### D.5    An Ergodic Theorem for non-homogeneous Markov Chains

We start with some definitions and a technical result. Let $\{P_\pi, \pi \in \Pi\}$ be a collection of Markov transition kernels on the state-action space $\mathcal{Z}$, indexed by policies $\pi \in \Pi$. For any Markov transition kernel $P$, bounded function $f$ and probability distribution $\mu$, we define:

$$Pf(z) \triangleq \sum_{z' \in \mathcal{Z}} P(z, z')f(z') \quad \text{and} \quad \mu P(z) \triangleq \sum_{z' \in \mathcal{Z}} \mu(z')P(z', z).$$

For a measure $\mu$ and a function $f$, $\mu(f) = \mathbb{E}_{X \sim \mu}[f(X)]$ denotes the mean of $f$ w.r.t. $\mu$. Finally, for two policies $\pi$ and $\pi'$ we define $D(\pi, \pi') \triangleq \|P_\pi - P_{\pi'}\|_\infty = \max_{z \in \mathcal{Z}} \|P_\pi(z, .) - P_{\pi'}(z, .)\|_1$.

We consider a $\mathcal{Z} \times \Pi$-valued process $\{(z_t, \pi_t), t \geq 1\}$ such that $(z_t, \pi_t)$ is $\mathcal{F}_t$-adapted and for any bounded measurable function $f$:

$$\mathbb{E}[f(z_{t+1})|\mathcal{F}_t] = P_{\pi_t}f(z_t).$$

The next result is adapted from [FMP11]. There the authors prove an ergodic theorem for adaptive Markov Chain Monte-Carlo (MCMC) algorithms with a general state space and a parameter-dependent function. For the sake of self-containedness, we include here the proof of their result in the simple case of finite state space chains with a function that does not depend on the policy $\pi_t$.

**Proposition 12.** *(Corollary 2.9, [FMP11]) Assume that:*

*(B1)* $\forall t \geq 1$, $P_t$ *is ergodic. We denote by $\omega_t$ its stationary distribution.*

*(B2) There exists an ergodic kernel $P$ such that $\|P_t - P\|_\infty \xrightarrow[t \to \infty]{} 0$ almost surely.*

*(B3) There exists two constants $C_t$ and $\rho_t$ such that for all $n \geq 1$, $\|P_t^n - W_t\|_\infty \leq C_t\rho_t^n$,*
        *where $W_t$ is a rank-one matrix whose rows are equal to $\omega_t^\mathsf{T}$.*

*(B4) Denote by $L_t \triangleq C_t(1 - \rho_t)^{-1}$. Then $\limsup_{t \to \infty} L_t < \infty$ almost surely. (UNIFORM ERGODICITY)*

*(B5)* $D(\pi_{t+1}, \pi_t) \xrightarrow[t \to \infty]{} 0$ *almost surely. (STABILITY)*

*Finally, denote by $\omega^\star$ the stationary distribution of $P$. Then for any bounded non-negative function $f : \mathcal{Z} \to \mathbb{R}^+$ we have:*

$$\frac{\sum_{k=1}^{t} f(z_k)}{t} \xrightarrow[t\to\infty]{} \omega^\star(f)$$

*almost surely.*

*Proof.* Consider the difference

$$D = \frac{\sum_{k=1}^{t} f(z_k)}{t} - \omega(f)$$

$$= \underbrace{\frac{f(z_1) - \omega^\star(f)}{t}}_{D_{1,t}} + \underbrace{\frac{\sum_{k=2}^{t} \left[f(z_k) - \omega_{k-1}(f)\right]}{t}}_{D_{2,t}} + \underbrace{\frac{\sum_{k=2}^{t} \left[\omega_{k-1}(f) - \omega^\star(f)\right]}{t}}_{D_{3,t}}. \tag{26}$$

$$\tag{27}$$

We clearly have:

$$|D_{1,t}| \leq \frac{\|f\|_\infty}{t} \xrightarrow[t\to\infty]{} 0. \tag{28}$$

Next, by Lemma 22 there exists a constant $\kappa_P$ (that only depends on $P$) such that: $\|\omega_k - \omega^\star\|_1 \leq \kappa_P \|P_k - P\|_\infty$. Therefore:

$$|D_{3,t}| \leq \kappa_P \|f\|_\infty \frac{\sum_{k=1}^{t-1} \|P_k - P\|_\infty}{t} \xrightarrow[t\to\infty]{} 0, \tag{29}$$

where the convergence to zero is due to assumption (B2). Now to bound $D_{2,t}$ we use the function $\widehat{f}_k$ solution to the Poisson equation $\left(\widehat{f}_k - P_k\widehat{f}_k\right)(.) = f(.) - \omega_k(f)$. By Lemma 23, $\widehat{f}_k(.) = \sum_{n\geq 0} P_k^n[f - \omega_k(f)](.)$ exists and is solution to the Poisson equation. Therefore we can rewrite $D_{2,t}$ as follows:

$$D_{2,t} = \frac{\sum_{k=2}^{t} \left[\widehat{f}_{k-1}(z_k) - P_{k-1}\widehat{f}_{k-1}(z_k)\right]}{t}$$

$$= M_t + C_t + R_t, \tag{30}$$

where

$$M_t \triangleq \frac{\sum_{k=2}^{t} \left[\widehat{f}_{k-1}(z_k) - P_{k-1}\widehat{f}_{k-1}(z_{k-1})\right]}{t},$$

$$C_t \triangleq \frac{\sum_{k=2}^{t} \left[P_k\widehat{f}_k(z_k) - P_{k-1}\widehat{f}_{k-1}(z_k)\right]}{t},$$

$$R_t \triangleq \frac{P_1\widehat{f}_1(z_1) - P_t\widehat{f}_t(z_t)}{t}.$$

**Bounding $M_t$:** Note that $S_t \triangleq tM_t$ is a martingale since $\mathbb{E}[\widehat{f}_{k-1}(z_k)|\mathcal{F}_{k-1}] = P_{k-1}\widehat{f}_{k-1}(z_{k-1})$. Furthermore, by Lemma 23:

$$|S_t - S_{t-1}| = |\widehat{f}_{t-1}(z_t) - P_{k-1}\widehat{f}_{t-1}(z_{t-1})|$$

$$\leq 2\left\|\widehat{f}_{t-1}\right\|_\infty$$

$$\leq 2L_{t-1}\|f\|_\infty.$$

In particular, this implies that:

$$\sum_{k=2}^{\infty} \frac{\mathbb{E}[|S_k - S_{k-1}|^2 \, |\mathcal{F}_{k-1}]}{k^2} \leq \sum_{k=2}^{\infty} \frac{4\,\|f\|_\infty^2 \, L_{k-1}^2}{k^2} < \infty$$

where the convergence of the series is due to (B4). (Theorem 2.18 in [HH80]) then implies that $M_t \underset{t\to\infty}{\longrightarrow} 0$ almost surely.

**Bounding $C_t$:**    Using Lemma 23, we have:

$$|C_t| \leq \|f\|_\infty \frac{\sum_{k=2}^{t} L_k \left[ \|\omega_k - \omega_{k-1}\|_1 + L_{k-1} D(\pi_k, \pi_{k-1}) \right]}{t}$$

$$\leq \|f\|_\infty \frac{\sum_{k=2}^{t} L_k \left[ \|\omega_k - \omega^\star\|_1 + \|\omega^\star - \omega_{k-1}\|_1 + L_{k-1} D(\pi_k, \pi_{k-1}) \right]}{t}$$

$$\leq \|f\|_\infty \left[ \kappa_P \frac{L_2 \|P_1 - P\|_\infty + L_t \|P_t - P\|_\infty + \sum_{k=2}^{t-1} (L_k + L_{k+1}) \|P_k - P\|_\infty}{t} \right.$$

$$\left. + \frac{\sum_{k=2}^{t} L_k L_{k-1} D(\pi_k, \pi_{k-1})}{t} \right] \underset{t\to\infty}{\longrightarrow} 0, \tag{31}$$

where the third line comes from applying Lemma 22 and the convergence to zero is due to assumptions (B2)-(B4)-(B5).

**Bounding $R_t$:**    Finally, by Lemma 23 we have:

$$|R_t| \leq \frac{\left\|\widehat{f}_1\right\|_\infty + \left\|\widehat{f}_t\right\|_\infty}{t}$$

$$\leq \frac{\|f\|_\infty (L_1 + L_t)}{t} \underset{t\to\infty}{\longrightarrow} 0, \tag{32}$$

where the convergence to zero is due to Assumption (B4). Summing up the inequalities (28-32) gives the result. □

### D.6    Application to C-Navigation: proof of Theorem 7

**We will now prove that C-Navigation verifies the assumptions (B1-5).**    The same can be proved for D-Navigation by replacing $\overline{\pi}_t^o$ with $\pi_t^o$. Theorem 7 follows immediately by applying Proposition 12 for the functions $f(\tilde{z}) = \mathbb{1}\{\tilde{z} = z\}$, where $z$ is any fixed state-action pair.

**(B1):**    This is a direct consequence of the fact that $P_{\pi_u}$ is ergodic (due to Assumptions 1 and 2) which implies by construction that $P_t$ is also ergodic.

**(B2):**    By Lemma 4 we have: $N_{sa}(t) \xrightarrow{a.s} \infty$ for all $(s, a)$. Hence $\widehat{\mathcal{M}} \xrightarrow{a.s} \mathcal{M}$ and by continuity: $\pi^o(\widehat{\mathcal{M}}) \xrightarrow{a.s} \pi^o(\mathcal{M})$, which implies that:

$$P_t \xrightarrow{a.s} P_{\pi^o}. \tag{33}$$

**(B3):**    By Lemma 13, $P_t$ satisfies (B3) for $C_t = 2\theta(\varepsilon_t, \overline{\pi}_t^o, \omega_t)^{-1}$ and $\rho_t = \theta(\varepsilon_t, \overline{\pi}_t^o, \omega_t)^{1/r}$.

**(B4):**    We have:

$$\sigma(\varepsilon_t, \overline{\pi}_t^o, \omega_t) = \left( \varepsilon_t^r + \left[ (1 - \varepsilon_t) A \min_{s,a} \overline{\pi}_t^o(a|s) \right]^r \right) \sigma_u \left( \min_z \frac{\omega_u(z)}{\omega_t(z)} \right)$$

$$\xrightarrow{a.s} \left( A \min_{s,a} \pi^o(a|s) \right) \sigma_u \min_z \frac{\omega_u(z)}{\omega^\star(z)} \triangleq \sigma_o. \tag{34}$$

Note that $\sigma_o > 0$ since $\omega_u > 0$ (ergodicity of $P_{\pi_u}$), $\omega^\star < 1$ and $\pi^o > 0$ entry-wise. Similarly, it is trivial that $\sigma_o < 1$ since $A \min_{s,a} \pi^o(a|s) < 1$, $\min_z \frac{\omega_u(z)}{\omega^\star(z)} < 1$ and $\sigma_u < 1$. Therefore $\theta(\varepsilon_t, \overline{\pi_t^o}, \omega_t) = 1 - \sigma(\varepsilon_t, \overline{\pi_t^o}, \omega_t) \xrightarrow{a.s} 1 - \sigma_o \triangleq \theta_o \in (0,1)$ and:

$$\limsup_{t \to \infty} L_t = \limsup_{t \to \infty} C_t(1 - \rho_t)^{-1}$$

$$= \limsup_{t \to \infty} \frac{2}{\theta(\varepsilon_t, \overline{\pi_t^o}, \omega_t)\big[1 - \theta(\varepsilon_t, \overline{\pi_t^o}, \omega_t)^{1/r}\big]}$$

$$= \frac{2}{\theta_o\big[1 - \theta_o^{\frac{1}{r}}\big]} < \infty. \tag{35}$$

**(B5):** We have: $\big(P_{t+1} - P_t\big)(s, s') = \sum_a [\pi_{t+1}(a|s) - \pi_t(a|s)]p_{\mathcal{M}}(s'|s, a)$. Hence: $\|P_{t+1} - P_t\|_\infty \le \|\pi_{t+1} - \pi_t\|_\infty$, where $\pi_{t+1}$ and $\pi_t$ are viewed as vectors. On the other hand:

$$\pi_{t+1} - \pi_t = (\varepsilon_t - \varepsilon_{t+1})(\overline{\pi_t^o} - \pi_u) + (1 - \varepsilon_t)(\overline{\pi_{t+1}^o} - \overline{\pi_t^o})$$

$$= (\varepsilon_t - \varepsilon_{t+1})(\overline{\pi_t^o} - \pi_u) + (1 - \varepsilon_t)(\frac{t \times \overline{\pi_t^o} + \pi^o(\widehat{\mathcal{M}}_{t+1})}{t+1} - \overline{\pi_t^o})$$

$$= (\varepsilon_t - \varepsilon_{t+1})(\overline{\pi_t^o} - \pi_u) + (1 - \varepsilon_t)\frac{\pi^o(\widehat{\mathcal{M}}_{t+1}) - \overline{\pi_t^o}}{t+1}$$

Therefore

$$\|P_{t+1} - P_t\|_\infty \le \|\pi_{t+1} - \pi_t\|_\infty$$

$$\le (\varepsilon_t - \varepsilon_{t+1}) + \frac{1}{t+1} \xrightarrow[t \to \infty]{} 0.$$

For D-Navigation, we get in a similar fashion:

$$\|P_{t+1} - P_t\|_\infty \le \|\pi_{t+1} - \pi_t\|_\infty$$

$$\le (\varepsilon_t - \varepsilon_{t+1}) + (1 - \varepsilon_t)\left\|\pi_{t+1}^o - \pi_t^o\right\|_\infty \xrightarrow[t \to \infty]{} 0.$$

### D.7 Geometric ergodicity of the sampling rules

Since $P_{\pi_u}$ is ergodic, there exists $r > 0$ such that $P_{\pi_u}^r(z, z') > 0$ for all $z, z'$ (Proposition 1.7, [LPW06]). Thus we define

$$r = \min\{\ell \ge 1 : \forall (z, z') \in \mathcal{Z}^2, \ P_{\pi_u}^\ell(z, z') > 0\}, \tag{36}$$

$$\sigma_u \triangleq \min_{z,z'} \frac{P_{\pi_u}^r(z, z')}{\omega_u(z')}, \tag{37}$$

where $\omega_u$ is the stationary distribution of $P_{\pi_u}$.

**Lemma 13.** *Let $\pi_t^o \triangleq \pi^o(\widehat{\mathcal{M}}_t)$ (resp. $\overline{\pi_t^o} \triangleq \sum_{j=1}^t \pi^o(\widehat{\mathcal{M}}_j)/t$) denote the oracle policy of $\widehat{\mathcal{M}}_t$ (resp. the Cesaro-mean of oracle policies up to time $t$). Further define*

$$\sigma(\varepsilon, \pi, \omega) \triangleq \left(\varepsilon^r + \big[(1 - \varepsilon)A \min_{s,a} \pi(a|s)\big]^r\right)\sigma_u\left(\min_z \frac{\omega_u(z)}{\omega(z)}\right),$$

$$\theta(\varepsilon, \pi, \omega) \triangleq 1 - \sigma(\varepsilon, \pi, \omega),$$

$$\mathcal{L}(\varepsilon, \pi, \omega) \triangleq \frac{2}{\theta(\varepsilon, \pi, \omega)\big[1 - \theta(\varepsilon, \pi, \omega)^{1/r}\big]}.$$

*Then for D-Navigation (resp. C-Navigation) we have:*

$$\forall n \ge 1, \ \|P_t^n - W_t\|_\infty \le C_t \rho_t^n$$

*where $C_t = 2\theta(\varepsilon_t, \pi_t^o, \omega_t)^{-1}$ and $\rho_t = \theta(\varepsilon_t, \pi_t^o, \omega_t)^{1/r}$ (resp. $C_t = 2\theta(\varepsilon_t, \overline{\pi_t^o}, \omega_t)^{-1}$ and $\rho_t = \theta(\varepsilon_t, \overline{\pi_t^o}, \omega_t)^{1/r}$). In particular $L_t \triangleq C_t(1 - \rho_t)^{-1} = \mathcal{L}(\varepsilon_t, \pi_t^o, \omega_t)$ (resp. $L_t = C_t(1 - \rho_t)^{-1} = \mathcal{L}(\varepsilon_t, \overline{\pi_t^o}, \omega_t)$).*

*Proof.* We only prove the lemma for C-Navigation. The statement for D-Navigation can be proved in the same way. Recall that: $P_t = \varepsilon_t P_{\pi_u} + (1 - \varepsilon_t) P_{\overline{\pi_t^o}}$. Therefore:

$$\forall (z, z'), \quad P_t^r(z, z') \geq [\varepsilon_t^r P_{\pi_u}^r + (1 - \varepsilon_t)^r P_{\overline{\pi_t^o}}^r](z, z')$$

$$\geq \left( \varepsilon_t^r + \left[ (1 - \varepsilon_t) A \min_{s,a} \overline{\pi_t^o}(a|s) \right]^r \right) P_{\pi_u}^r(z, z')$$

$$\geq \left( \varepsilon_t^r + \left[ (1 - \varepsilon_t) A \min_{s,a} \overline{\pi_t^o}(a|s) \right]^r \right) \sigma_u \omega_u(z')$$

$$\geq \underbrace{\left( \varepsilon_t^r + \left[ (1 - \varepsilon_t) A \min_{s,a} \overline{\pi_t^o}(a|s) \right]^r \right) \sigma_u \left( \min_z \frac{\omega_u(z)}{\omega_t(z)} \right)}_{\sigma_t} \omega_t(z')$$

$$= \sigma(\varepsilon_t, \overline{\pi_t^o}, \omega_t) \omega_t(z').$$

where the second and third inequalities comes from the fact that $P_{\overline{\pi_t^o}} \geq A \min_{s,a} \overline{\pi_t^o}(a|s) P_{\pi_u}$ entry-wise and from (37) respectively. Using Lemma 21 we conclude that or all $n \geq 1$:

$$\|P_t^n - W_t\|_\infty \leq 2\theta(\varepsilon_t, \overline{\pi_t^o}, \omega_t)^{\frac{n}{r} - 1}$$

where $\theta(\varepsilon_t, \overline{\pi_t^o}, \omega_t) = 1 - \sigma(\varepsilon_t, \overline{\pi_t^o}, \omega_t)$. Therefore $P_t$ satisfies $\|P_t^n - W_t\|_\infty \leq C_t \rho_t^n$ for $C_t = 2\theta(\varepsilon_t, \overline{\pi_t^o}, \omega_t)^{-1}$ and $\rho_t = \theta(\varepsilon_t, \overline{\pi_t^o}, \omega_t)^{1/r}$. $\square$

# E  Stopping rule

## E.1  Deviation inequality for KL divergences of rewards

We suppose that the reward distributions $q_{\mathcal{M}}(s, a)$ come from a one-dimensional exponential family and can therefore be parametrized by their respective means $r_{\mathcal{M}}(s, a)$. Furthermore, for any $t$ such that $N_{sa}(t) > 0$, we let $\widehat{q}_{s,a}(t)$ denote the distribution belonging to the same exponential family, whose mean is the empirical average $\widehat{r}_t(s, a) = \frac{\sum\limits_{k=1}^t R_k \mathbb{1}\{(s_t, a_t) = (s,a)\}}{N_{sa}(t)}$. For $x \geq 1$, define the function $h(x) = x - \log(x)$ and its inverse $h^{-1}(x)$. Further define the function $\tilde{h} : \mathbb{R}^+ \longrightarrow \mathbb{R}$ by:

$$\tilde{h}(x) = \begin{cases} h^{-1}(x) \exp(1/h^{-1}(x)) & \text{if } x \geq h^{-1}(1/\ln(3/2)), \\ \frac{3}{2}\left[x - \log(\log(3/2)\right] & \text{otherwise.} \end{cases}$$

Finally let

$$\varphi(x) = 2\tilde{h}\left( \frac{h^{-1}(1 + x) + \log(2\Gamma(2))}{2} \right),$$

where $\Gamma(2) = \sum_{n=1}^\infty 1/n^2$. Now we recall a deviation inequality from [KK18], which we use for the empirical KL divergence of rewards.

**Lemma 14.** *(Theorem 14, [KK18]) Define the threshold $\beta_r(t, \delta) \triangleq SA\varphi\big( \log(1/\delta)/SA \big) + 3\sum\limits_{s,a} \log\big[ 1 + \log(N_{sa}(t)) \big]$. Then for all $\delta \in (0, 1)$:*

$$\mathbb{P}\left( \exists t \geq 1, \sum_{(s,a) \in \mathcal{Z}} N_{sa}(t) \, \mathrm{KL}(\widehat{q}_{s,a}(t), q_{\mathcal{M}}(s, a)) > \beta_p(t, \delta) \right) \leq \delta.$$

**Remark 5.** *One can easily see that $\varphi(x) \underset{x \to \infty}{\sim} x$.*

## E.2 Deviation inequality for KL divergences of transitions

Our second deviation inequality is adapted from Proposition 1 in [JKM+20]. There the authors derive a deviation inequality for a *single KL divergence* of a multinomial distribution. In order to get a deviation inequality of a *sum of KL divergences*, we modified their proof by considering the product over state-action pairs of the martingales they used. For the sake of self-containedness, we include the proof below. $\widehat{p}_{sa}(t)$ is defined as the categorical distribution with a vector of probabilities $q$ satisfying:

$$\forall s' \in \mathcal{S}, \ q_{s'} = \begin{cases} \dfrac{\sum_{k=0}^{t-1} \mathbb{1}\{(s_k, a_k, s_{k+1}) = (s, a, s')\}}{N_{sa}(t)} & \text{if } N_{sa}(t) \neq 0, \\[2mm] 1/A & \text{otherwise.} \end{cases}$$

**Lemma 15.** *(Proposition 1, [JKM+20]) Define the threshold* $\beta_p(t,\delta) \triangleq \log(1/\delta) + (S - 1)\sum_{s,a} \log\left(e\left[1 + N_{sa}(t)/(S-1)\right]\right)$. *Then for all* $\delta \in (0,1)$ *we have:*

$$\mathbb{P}\left(\exists t \geq 1, \ \sum_{(s,a)\in\mathcal{Z}} N_{sa}(t) \, \mathrm{KL}(\widehat{p}_{sa}(t), \, p_{\mathcal{M}}(s,a)) > \beta_p(t,\delta)\right) \leq \delta,$$

*with the convention that* $N_{sa}(t) \, \mathrm{KL}(\widehat{p}_{sa}(t), \, p_{\mathcal{M}}(s,a)) = 0$ *whenever* $N_{sa}(t) = 0$.

*Proof.* We begin with a few notations. For any vector $\lambda \in \mathbb{R}^{S-1} \times \{0\}$ and any element of the simplex $p \in \Sigma_{S-1}$, we denote $< \lambda, p > \triangleq \sum_{i=1}^{S-1} \lambda_i p_i$. We define the log-partition function of a discrete distribution $p$ supported over $\{1, \ldots, S\}$ by:

$$\forall \lambda \in \mathbb{R}^{S-1} \times \{0\}, \ \phi_p(\lambda) \triangleq \log\left(p_S + \sum_{i=1}^{S-1} p_i e_i^{\lambda}\right).$$

We use the shorthand and let $\phi_{sa}(\lambda) \triangleq \phi_{p_{sa}}(\lambda)$. For $N \in \mathbb{N}^*$ and $x \in \{0, \ldots, N\}^k$ such that $\sum_{i=1}^k x_i = N$ the binomial coefficient is defined as: $\binom{N}{x} \triangleq \frac{N!}{\prod_{i=1}^k x_i!}$. Finally $H(p) = \sum_{i=1}^{S} p_i \log(1/p_i)$ is the Shannon entropy of distribution $p$.

**Building a convenient mixture martingale for every state-action pair:** Following [JKM+20], we define for every integer $t$:

$$M_t^{\lambda}(s,a) = \exp\left(N_{sa}(t)\big(< \lambda, \widehat{p}_{sa}(t) > -\phi_{sa}(\lambda)\big)\right). \tag{38}$$

The sequence $(M_t^{\lambda}(s,a))_t$ is an $(\mathcal{F}_t)_t$-martingale since:

$$\mathbb{E}[M_t^{\lambda}(s,a)|\mathcal{F}_{t-1}, \ (s_t, a_t) = (s,a)]$$
$$= \mathbb{E}\left[\exp\left(N_{sa}(t)\big[< \lambda, \widehat{p}_{sa}(t) > -\phi_{sa}(\lambda)\big]\right)\Big| \mathcal{F}_{t-1}, \ (s_t, a_t) = (s,a)\right]$$
$$= \mathbb{E}_{X\sim p_{sa}}\left[\exp\left((N_{sa}(t-1)+1)\big(< \lambda, \frac{N_{sa}(t-1)\widehat{p}_{sa}(t-1) + X}{N_{sa}(t-1)+1} > -\phi_{sa}(\lambda)\big)\right)\Big| \mathcal{F}_{t-1}\right]$$
$$= \mathbb{E}_{X\sim p_{sa}}\left[M_{t-1}^{\lambda}(s,a) \exp\left(< \lambda, X > -\phi_{sa}(\lambda)\right)\Big| \mathcal{F}_{t-1}\right] = M_{t-1}^{\lambda}(s,a).$$

The same holds trivially when $(s_t, a_t) \neq (s,a)$. Now, we define the mixture martingale defined by the family of priors $\lambda_q = \nabla \phi_{sa}^{-1}(q)$ where $q \sim \mathcal{D}\mathrm{ir}(1, \ldots, 1)$ follows a Dirichlet distribution with

parameters $(1, \ldots, 1)$:

$$M_t(s,a) = \int M_t^{\lambda_q}(s,a) \frac{\Gamma(S)}{\prod_{i=1}^{S} \Gamma(1)} \prod_{i=1}^{S} q_i dq$$

$$= \int e^{N_{sa}(t)\left(KL(\widehat{p}_{sa}(t), p_{sa}) - KL(\widehat{p}_{sa}(t), q)\right)} (S-1)! \prod_{i=1}^{S} q_i dq$$

$$= \exp\left(N_{sa}(t)\left(KL(\widehat{p}_{sa}(t), p_{sa}) + H(\widehat{p}_{sa}(t))\right)\right)(S-1)! \int \prod_{i=1}^{S} q_i^{1+N_{sa}(t)\widehat{p}_{sa,i}(t)} dq$$

$$= \exp\left(N_{sa}(t)\left[KL(\widehat{p}_{sa}(t), p_{sa}) + H(\widehat{p}_{sa}(t))\right]\right) \frac{(S-1)! \prod_{i=1}^{S} \Gamma\left(1 + N_{sa}(t)\widehat{p}_{sa,i}(t)\right)}{\Gamma(N_{sa}(t) + S)}$$

$$= \exp\left(N_{sa}(t)\left[KL(\widehat{p}_{sa}(t), p_{sa}) + H(\widehat{p}_{sa}(t))\right]\right) \frac{(S-1)! \prod_{i=1}^{S} \left(N_{sa}(t)\widehat{p}_{sa,i}(t)\right)!}{(N_{sa}(t) + S - 1)!}$$

$$= \exp\left(N_{sa}(t)\left[KL(\widehat{p}_{sa}(t), p_{sa}) + H(\widehat{p}_{sa}(t))\right]\right) \frac{\prod_{i=1}^{S} \left(N_{sa}(t)\widehat{p}_{sa,i}(t)\right)!}{N_{sa}(t)!} \frac{(S-1)! N_{sa}(t)!}{(N_{sa}(t) + S - 1)!}$$

$$= \exp\left(N_{sa}(t)\left[KL(\widehat{p}_{sa}(t), p_{sa}) + H(\widehat{p}_{sa}(t))\right]\right) \frac{1}{\binom{N_{sa}(t)+S-1}{S-1}} \frac{1}{\binom{N_{sa}(t)}{N_{sa}(t)\widehat{p}_{sa}(t)}},$$

where in the second inequality we used Lemma 16 and $\widehat{p}_{sa,i}(t)$ denotes the i-th component of $\widehat{p}_{sa}(t)$. Now using Lemma 17, we upper bound the binomial coefficients which leads to:

$$M_t(s,a) \geq \exp\left( N_{sa}(t)\left[KL(\widehat{p}_{sa}(t), p_{sa}) + H(\widehat{p}_{sa}(t))\right] - N_{sa}(t)H(\widehat{p}_{sa}(t)) \right.$$
$$\left. - (N_{sa}(t) + S - 1)H(S - 1/(N_{sa}(t) + S - 1)) \right)$$
$$= \exp\left( N_{sa}(t)KL(\widehat{p}_{sa}(t), p_{sa}) - (N_{sa}(t) + S - 1)H(S - 1/(N_{sa}(t) + S - 1)) \right).$$

**The product martingale:** Taking the product over all state-action pairs we get:

$$M_t \triangleq \prod_{(s,a)\in\mathcal{Z}} M_t(s,a)$$
$$\geq \exp\left( \sum_{s,a} N_{sa}(t)KL(\widehat{p}_{sa}(t), p_{sa}) - \sum_{s,a}(N_{sa}(t) + S - 1)H(S - 1/(N_{sa}(t) + S - 1)) \right). \tag{39}$$

Next, using that $\log(1 + x) \leq x$ we get:

$$(N_{sa}(t) + S - 1)H(S - 1/(N_{sa}(t) + S - 1)) = (S-1)\log(1 + N_{sa}(t)/(S-1))$$
$$+ N_{sa}(t)\log(1 + (S-1)/N_{sa}(t))$$
$$\leq (S-1)\log(1 + N_{sa}(t)/(S-1)) + (S-1)$$
$$= (S-1)\log\left(e\left[1 + N_{sa}(t)/(S-1)\right]\right).$$

Hence (39) becomes:

$$M_t \geq \exp\left( \sum_{s,a} N_{sa}(t)KL(\widehat{p}_{sa}(t), p_{sa}) - (S-1)\sum_{s,a}\log\left(e\left[1 + N_{sa}(t)/(S-1)\right]\right) \right). \tag{40}$$

Now, we show that $M_t$ is a martingale. For any fixed pair $(s, a)$ we have:

$$\mathbb{E}[M_t|\mathcal{F}_{t-1},\ (s_t, a_t) = (s, a)] = \mathbb{E}[M_t(s, a) \prod_{(s', a') \neq (s, a)} M_t(s', a')|\mathcal{F}_{t-1},\ (s_t, a_t) = (s, a)]$$

$$= \mathbb{E}[M_t(s, a) \prod_{(s', a') \neq (s, a)} M_{t-1}(s', a')|\mathcal{F}_{t-1}]$$

$$= \mathbb{E}[M_t(s, a)|\mathcal{F}_{t-1}] \times \prod_{(s', a') \neq (s, a)} M_{t-1}(s', a')$$

$$= M_{t-1},$$

where the third equality is because $M_t(s, a)$ and $\left(M_{t-1}(s', a')\right)_{(s', a') \neq (s, a)}$ are independent conditionally on $\mathcal{F}_{t-1}$. Finally, using the tower rule we get:

$$\mathbb{E}[M_t|\mathcal{F}_{t-1}] = \mathbb{E}\left[\mathbb{E}[M_t|\mathcal{F}_{t-1},\ (s_t, a_t)]\middle|\mathcal{F}_{t-1}\right] = \mathbb{E}[M_{t-1}|\mathcal{F}_{t-1}] = M_{t-1}.$$

Hence $M_t$ is a martingale. Thanks to Doob's maximal inequality we have:

$$\mathbb{P}\left(\exists t \geq 0,\ M_t > 1/\delta\right) \leq \delta\mathbb{E}[M_0] = \delta.$$

In view of (40), we conclude that for $\beta_p(t, \delta) = \log(1/\delta) + (S-1)\sum_{s,a} \log\left(e\left[1 + N_{sa}(t)/(S-1)\right]\right)$ we have:

$$\mathbb{P}\left(\exists t \geq 1,\ \sum_{s,a} N_{sa}(t)KL(\widehat{p}_{sa}(t), p_{sa}) > \beta_p(t, \delta)\right) \leq \delta.$$

$\square$

**Lemma 16.** *(Lemma 3 in [JKM$^+$20]) For $q, p$ in $\Sigma_m$ the simplex of dimension $(m-1)$ and $\lambda \in \mathbb{R}^{m-1}$, we have:*

$$< \lambda, q > -\phi_p(\lambda) = KL(q, p) - KL(q, p^\lambda)$$

*where $p^\lambda = \nabla\phi_p(\lambda)$.*

**Lemma 17.** *(Theorem 11.1.3, [CT06]) Let $N \in \mathbb{N}^*$, and $x \in \{0, \ldots, N\}^k$ such that $\sum_{i=1}^{k} x_i = N$ then:*

$$\binom{N}{x} = \frac{N!}{\prod_{i=1}^{k} x_i!} \leq e^{NH(x/N)}$$

*where $H(x/N)$ is the Shannon entropy of the discrete distribution over $\{1, \ldots, k\}$ with vector of probabilities $(\frac{x_i}{N})_{1 \leq i \leq k}$.*

### E.3 Correctness of the stopping rule

*Proof.* Using Lemma 3 and equations (11-12) in the second and third lines respectively, we get:

$$\mathbb{P}(\widehat{\pi}^*_\tau \neq \pi^*, \tau_\delta < \infty) = \mathbb{P}\left(\exists t \geq 1,\ t\,U(\widehat{\mathcal{M}}_t, \boldsymbol{N}(t)/t)^{-1} \geq \beta_r(t,\delta/2) + \beta_p(t,\delta/2), \widehat{\pi}^*_t \neq \pi^*\right)$$

$$\leq \mathbb{P}\left(\exists t \geq 1,\ t\,T(\widehat{\mathcal{M}}_t, \boldsymbol{N}(t)/t)^{-1} \geq \beta_r(t,\delta/2) + \beta_p(t,\delta/2), \mathcal{M} \in \mathrm{Alt}(\widehat{\mathcal{M}}_t)\right)$$

$$= \mathbb{P}\left(\exists t \geq 1,\ \inf_{\mathcal{M}' \in \mathrm{Alt}(\widehat{\mathcal{M}}_t)} \sum_{(s,a)\in\mathcal{Z}} N_{sa}(t)\big[\,\mathrm{KL}(\widehat{q}_{s,a}(t), q_{\mathcal{M}'}(s,a)) + \mathrm{KL}(\widehat{p}_{sa}(t), p_{\mathcal{M}'}(s,a))\big]\right.$$

$$\left. \geq \beta_r(t,\delta/2) + \beta_p(t,\delta/2), \mathcal{M} \in \mathrm{Alt}(\widehat{\mathcal{M}}_t)\right)$$

$$\leq \mathbb{P}\left(\exists t \geq 1,\ \sum_{(s,a)\in\mathcal{Z}} N_{sa}(t)\big[\,\mathrm{KL}(\widehat{q}_{s,a}(t), q_{\mathcal{M}}(s,a)) + \mathrm{KL}(\widehat{p}_{sa}(t), p_{\mathcal{M}}(s,a))\big]\right.$$

$$\left. \geq \beta_r(t,\delta/2) + \beta_p(t,\delta/2)\right)$$

$$= \mathbb{P}\left(\exists t \geq 1,\ \sum_{(s,a)\in\mathcal{Z}} N_{sa}(t)\,\mathrm{KL}(\widehat{q}_{s,a}(t), q_{\mathcal{M}}(s,a)) \geq \beta_r(t,\delta/2)\right)$$

$$+ \mathbb{P}\left(\exists t \geq 1,\ \sum_{(s,a)\in\mathcal{Z}} N_{sa}(t)\,\mathrm{KL}(\widehat{p}_{sa}(t), p_{\mathcal{M}}(s,a)) \geq \beta_p(t,\delta/2)\right)$$

$$\leq \delta/2 + \delta/2 = \delta$$

where the last inequality is due to Lemmas 14 and 15. □

## F  Sample complexity upper bound

### F.1  Almost sure upper bound: proof of Theorem 6 *(i)*

*Proof.* Consider the event $\mathcal{E} = \left(\forall (s,a) \in \mathcal{Z},\ \lim_{t\to\infty} \frac{N_{sa}(t)}{t} = \omega^\star_{s,a},\ \widehat{\mathcal{M}}_t \to \mathcal{M}\right)$. By Lemma 4 and Theorem 7, we have $\mathbb{P}(\mathcal{E}) = 1$. We will prove that under $\mathcal{E}$, $\limsup_{\delta\to 0} \frac{\tau_\delta}{\log(1/\delta)} \leq 2U_o(\mathcal{M})$.
Fix $\eta > 0$. There exits $t_\eta$ such that for all $t \geq t_\eta$:

$$U(\widehat{\mathcal{M}}_t, \boldsymbol{N}(t)/t)^{-1} \geq (1-\eta)U(\mathcal{M}, \omega^\star)^{-1} \tag{41}$$

$$\beta_p(t,\delta/2) \leq \log(1/\delta) + \eta U(\mathcal{M}, \omega^\star)^{-1} t \tag{42}$$

$$\beta_r(t,\delta/2) \leq SA\,\varphi\big(\log(1/\delta)/SA\big) + \eta U(\mathcal{M}, \omega^\star)^{-1} t \tag{43}$$

where the last two inequalities come from the fact that both the thresholds satisfy $\beta(t,\delta/2) = \mathcal{O}\big(\log(t)\big) = o(t)$. Combining the inequalities above with the definition of $\tau_\delta$, we get:

$$\tau_\delta \leq \inf\left\{ t \geq t_\eta, (1-3\eta)tU(\mathcal{M}, \omega^\star)^{-1} \geq \log(1/\delta) + SA\,\varphi\big(\log(1/\delta)/SA\big)\right\}$$

$$= \max\left( t_\eta, \frac{\Big[log(1/\delta) + SA\,\varphi\big(\log(1/\delta)/SA\big)\Big]U(\mathcal{M}, \omega^\star)}{1-3\eta}\right).$$

Since $\varphi(x) \underset{\infty}{\sim} x$, then the last inequality implies that $\limsup_{\delta\to 0} \frac{\tau_\delta}{\log(1/\delta)} \leq \frac{2U(\mathcal{M},\omega^\star)}{1-3\eta}$. Taking the limit when $\eta$ goes to zero finishes the proof. □

## F.2 Upper bound in expectation: proof of Theorem 6 *(ii)*

*Proof.* We start by defining the semi-distance between MDPs:

$$\|\mathcal{M} - \mathcal{M}'\| = \max_{s,a} \max \left( |r_{\mathcal{M}}(s,a) - r_{\mathcal{M}'}(s,a)|, \|p_{\mathcal{M}}(.|s,a) - p_{\mathcal{M}'}(.|s,a)\|_1 \right).$$

Now for $\xi > 0$, by continuity of $\mathcal{M} \to \pi^o(\mathcal{M})$[17] there exists $\rho(\xi) \leq \xi$ such that:

$$\forall \mathcal{M}' \in \mathcal{B}(\mathcal{M}, \rho(\xi)), \ \|\pi^o(\mathcal{M}') - \pi^o(\mathcal{M})\|_\infty \leq \xi$$

where $\mathcal{B}(\mathcal{M}, \rho) = \{\mathcal{M}' : \|\mathcal{M}' - \mathcal{M}\| \leq \rho\}$. For $T \geq 1$, consider the concentration events[18]:

$$\mathcal{C}_T^1(\xi) \triangleq \bigcap_{t=T^{1/4}}^{T} \left( \widehat{\mathcal{M}}_t \in \mathcal{B}(\mathcal{M}, \rho(\xi)) \right).$$

$$\mathcal{C}_T^2(\xi) \triangleq \bigcap_{t=T^{3/4}}^{T} \left( |\mathbf{N}(t)/t - \omega^\star| \leq K_\xi \xi \right)$$

where $K_\xi$ is a mapping defined in Proposition 19. We will upper bound the stopping time of MDP-NaS under $\mathcal{C}_T^1(\xi) \cap \mathcal{C}_T^2(\xi)$. Define:

$$U(\mathcal{M}, \omega^\star, \xi) \triangleq \sup_{\substack{\mathcal{M}' \in \mathcal{B}(\mathcal{M}, \rho(\xi)) \\ \|\omega' - \omega^\star\|_\infty \leq K_\xi \xi}} U(\mathcal{M}', \omega').$$

By Proposition 19, there exists $T_1(\xi)$ such that for all $T \geq T_1(\xi)$, conditionally on $\mathcal{C}_T^1(\xi)$, the event $\mathcal{C}_T^2(\xi)$ occurs with high probability. For $T \geq T_1(\xi)$ we have:

$$\forall t \in [[T^{3/4}, T]], \ U\left(\widehat{\mathcal{M}}_t, \mathbf{N}(t)/t\right) \leq U(\mathcal{M}, \omega^\star, \xi). \tag{44}$$

Furthermore, using the bound $N_{sa}(t) \leq t$ in the definitions of the thresholds $\beta_p$ and $\beta_r$ and the fact that $\log(t) \underset{t\to\infty}{=} o(t)$, we prove the existence of $T_2(\xi)$ such that for all $t \geq T_2(\xi)$:

$$\beta_p(t, \delta/2) \leq \log(1/\delta) + \xi U(\mathcal{M}, \omega^\star, \xi)^{-1} t \tag{45}$$

$$\beta_r(t, \delta/2) \leq SA \, \varphi\left( \log(1/\delta)/SA \right) + \xi U(\mathcal{M}, \omega^\star, \xi)^{-1} t. \tag{46}$$

Finally define:

$$T_3(\xi, \delta) \triangleq \frac{U(\mathcal{M}, \omega^\star, \xi) \left[ \log(1/\delta) + SA\varphi\left(\frac{\log(1/\delta)}{SA}\right) \right]}{(1 - 2\xi)}.$$

Using (44-46), we have for all $T \geq \max\left(T_1(\xi), T_2(\xi), T_3(\xi, \delta)\right)$ under $\mathcal{C}_T^1(\xi) \cap \mathcal{C}_T^2(\xi)$ the following holds:

$$T \times U\left(\widehat{\mathcal{M}}_T, \mathbf{N}(T)/T\right)^{-1} \geq \beta_p(T, \delta/2) + \beta_r(T, \delta/2).$$

In other words:

$$\forall T \geq \max\left(T_1(\xi), T_2(\xi), T_3(\xi, \delta)\right), \quad \mathcal{C}_T^1(\xi) \cap \mathcal{C}_T^2(\xi) \subset \left(\tau_\delta \leq T\right). \tag{47}$$

---

[17] As a consequence of Berge's Theorem, which gives the continuity of $\mathcal{M} \mapsto \boldsymbol{\omega}^\star(\mathcal{M})$.

[18] For simplicity and w.l.o.g, we consider that $T^{1/4}$ and $T^{3/4}$ are integers.

Therefore[19]:

$$\mathbb{E}[\tau_\delta] = \sum_{T=1}^{\infty} \mathbb{P}(\tau_\delta > T)$$

$$\leq \max\left(T_1(\xi), T_2(\xi), T_3(\xi,\delta)\right) + \sum_{T=\max(T_1,T_2,T_3)}^{\infty} \mathbb{P}(\tau_\delta > T)$$

$$\leq \max\left(T_1(\xi), T_2(\xi), T_3(\xi,\delta)\right) + \sum_{T=\max(T_1,T_2,T_3)}^{\infty} \mathbb{P}\left(\overline{\mathcal{C}_T^1(\xi)} \cup \overline{\mathcal{C}_T^2(\xi)}\right)$$

$$\leq \max\left(T_1(\xi), T_2(\xi), T_3(\xi,\delta)\right) + \sum_{T=1}^{\infty} \left[\mathbb{P}\left(\overline{\mathcal{C}_T^1(\xi)}\right) + \mathbb{P}\left(\overline{\mathcal{C}_T^2(\xi)}\Big|\mathcal{C}_T^1(\xi)\right)\right]$$

$$\leq \max\left(T_1(\xi), T_2(\xi), T_3(\xi,\delta)\right) + \sum_{T=1}^{\infty} \frac{1}{T^2} + BT\exp\left(-\frac{CT^{1/16}}{\sqrt{\log(1+SAT^2)}}\right)$$

$$+ \frac{2SA\exp(-T^{3/4}\xi^2)}{1-\exp(-\xi^2)},$$

where we used Lemma 18 and Proposition 19 in the last inequality. This implies that $\mathbb{E}[\tau_\delta]$ is finite and:

$$\limsup_{\delta\to 0} \frac{\mathbb{E}[\tau_\delta]}{\log(1/\delta)} \leq \limsup_{\delta\to 0} \frac{T_3(\xi,\delta)}{\log(1/\delta)}$$

$$= \frac{2U\left(\mathcal{M},\omega^\star,\xi\right)}{(1-2\xi)}$$

where we used $\varphi(x) \underset{\infty}{\sim} x + \log(x)$. Finally, we take the limit $\xi \to 0$. Since $\rho(\xi) \leq \xi$ and $\limsup_{\xi\to 0} K_\xi < \infty$ then $\limsup_{\xi\to 0} U\left(\mathcal{M},\omega^\star,\xi\right) = U\left(\mathcal{M},\omega^\star\right) = U_o(\mathcal{M})$ which finishes the proof. $\quad\square$

### F.3 Concentration of the empirical MDPs

**Lemma 18.** *Define the event* $\mathcal{C}_T^1(\xi) \triangleq \bigcap_{t=T^{1/4}}^{T} \left(\widehat{\mathcal{M}}_t \in \mathcal{B}(\mathcal{M}, \rho(\xi))\right)$. *Then there exists two positive constants* $B$ *and* $C$ *that only depend on* $\xi$ *and* $\mathcal{M}$ *such that:*

$$\forall T \geq 1, \ \mathbb{P}\left(\overline{\mathcal{C}_T^1(\xi)}\right) \leq \frac{1}{T^2} + BT\exp\left(-\frac{CT^{1/16}}{\sqrt{\log(1+SAT^2)}}\right).$$

*Proof.* For simplicity we will denote $\rho(\xi)$ by $\rho$. Consider the forced exploration event:

$$\mathcal{E}_T = \left(\forall(s,a) \in \mathcal{Z}, \ \forall t \geq 1, \ N_{sa}(t) \geq \left[\frac{t}{\lambda(T)}\right]^{1/4} - 1\right)$$

where $\lambda(T) \triangleq \frac{(m+1)^2}{\eta^2}\log^2(1+SAT^2)$ and $\eta$ is a parameter that only depends on $\mathcal{M}$. Applying Corollary 1 for $\alpha = \frac{1}{T^2}$, we get $\mathbb{P}(\mathcal{E}_T) \geq 1 - 1/T^2$. Therefore we have:

$$\mathbb{P}\left(\overline{\mathcal{C}_T^1(\xi)}\right) \leq \mathbb{P}(\overline{\mathcal{E}_T}) + \mathbb{P}(\overline{\mathcal{C}_T^1(\xi)} \cap \mathcal{E}_T)$$

$$\leq \frac{1}{T^2} + \mathbb{P}(\overline{\mathcal{C}_T^1(\xi)} \cap \mathcal{E}_T). \tag{48}$$

---

[19] $\overline{\mathcal{E}}$ denotes the complementary of event $\mathcal{E}$.

On the other hand:

$$\mathbb{P}(\overline{\mathcal{C}_T^1(\xi)} \cap \mathcal{E}_T) \leq \sum_{t=T^{1/4}}^{T} \mathbb{P}\left(\widehat{\mathcal{M}}_t \notin \mathcal{B}(\mathcal{M}, \rho(\xi)) \cap \mathcal{E}_T\right)$$

$$\leq \sum_{t=T^{1/4}}^{T} \sum_{s,a} \left[ \mathbb{P}\left( \left(\widehat{r}_t(s,a) - r(s,a) > \rho\right) \cap \mathcal{E}_T \right) \right.$$

$$+ \mathbb{P}\left( \left(\widehat{r}_t(s,a) - r(s,a) < -\rho\right) \cap \mathcal{E}_T \right)$$

$$+ \sum_{s'} \mathbb{P}\left( \left(\widehat{p}_t(s'|s,a) - p(s'|s,a) > \rho/S\right) \cap \mathcal{E}_T \right)$$

$$\left. + \mathbb{P}\left( \left(\widehat{p}_t(s'|s,a) - p(s'|s,a) < -\rho/S\right) \cap \mathcal{E}_T \right) \right]. \qquad (49)$$

Using a union bound and Chernoff-Hoeffding theorem respectively we get for $t \geq T^{1/4}$:

$$\mathbb{P}\left( \left(\widehat{p}_t(s'|s,a) - p(s'|s,a) > \rho/S\right) \cap \mathcal{E}_T \right)$$

$$\leq \mathbb{P}\left( \widehat{p}_t(s'|s,a) - p(s'|s,a) > \rho/S, \ N_{sa}(t) \geq \left[\frac{t}{\lambda(T)}\right]^{1/4} - 1 \right)$$

$$= \sum_{t'=\left[\frac{t}{\lambda(T)}\right]^{1/4}-1}^{t} \mathbb{P}\left( \widehat{p}_t(s'|s,a) - p(s'|s,a) > \rho/S, \ N_{sa}(t) = t' \right)$$

$$\leq \sum_{t'=\left[\frac{t}{\lambda(T)}\right]^{1/4}-1}^{t} \exp\left( -t' \cdot \mathrm{kl}\left(p(s'|s,a) + \rho/S, \ p(s'|s,a)\right) \right)$$

$$\leq \frac{\exp\left( -\left(\left[\frac{t}{\lambda(T)}\right]^{1/4} - 1\right) \mathrm{kl}\left(p(s'|s,a) + \rho/S, \ p(s'|s,a)\right) \right)}{1 - \exp\left( -\mathrm{kl}\left(p(s'|s,a) + \rho/S, \ p(s'|s,a)\right) \right)}$$

$$\leq \frac{\exp\left( -\left[\frac{T^{1/16}}{\lambda(T)^{1/4}} - 1\right] \mathrm{kl}\left(p(s'|s,a) + \rho/S, \ p(s'|s,a)\right) \right)}{1 - \exp\left( -\mathrm{kl}\left(p(s'|s,a) + \rho/S, \ p(s'|s,a)\right) \right)}. \qquad (50)$$

In a similar fashion we prove that:

$$\mathbb{P}\left( \left(\widehat{p}_t(s'|s,a) - p(s'|s,a) < -\rho/S\right) \cap \mathcal{E}_T \right)$$

$$\leq \frac{\exp\left( -\left[\frac{T^{1/16}}{\lambda(T)^{1/4}} - 1\right] \mathrm{kl}\left(p(s'|s,a) - \rho/S, \ p(s'|s,a)\right) \right)}{1 - \exp\left( -\mathrm{kl}\left(p(s'|s,a) - \rho/S, \ p(s'|s,a)\right) \right)}, \qquad (51)$$

$$\mathbb{P}\left( \left(\widehat{r}_t(s,a) - r(s,a) > \rho\right) \cap \mathcal{E}_T \right) \leq \frac{\exp\left( -\left[\frac{T^{1/16}}{\lambda(T)^{1/4}} - 1\right] \mathrm{kl}\left(r(s,a) + \rho, \ r(s,a)\right) \right)}{1 - \exp\left( -\mathrm{kl}\left(r(s,a) + \rho, \ r(s,a)\right) \right)}, \qquad (52)$$

$$\mathbb{P}\left( \left(\widehat{r}_t(s,a) - r(s,a) < -\rho\right) \cap \mathcal{E}_T \right) \leq \frac{\exp\left( -\left[\frac{T^{1/16}}{\lambda(T)^{1/4}} - 1\right] \mathrm{kl}\left(r(s,a) - \rho, \ r(s,a)\right) \right)}{1 - \exp\left( -\mathrm{kl}\left(r(s,a) - \rho, \ r(s,a)\right) \right)}. \qquad (53)$$

Thus, for the following choice of constants

$$C = \sqrt{\frac{\eta}{m+1}} \min_{s,a,s'} \Bigg( \mathrm{kl}\big(r(s,a) - \rho,\ r(s,a)\big),\ \mathrm{kl}\big(r(s,a) + \rho,\ r(s,a)\big),$$

$$\mathrm{kl}\big(p(s'|s,a) - \rho/S,\ p(s'|s,a)\big),\ \mathrm{kl}\big(p(s'|s,a) + \rho/S,\ p(s'|s,a)\big)\Bigg),$$

$$B = \sum_{s,a} \Bigg( \frac{\exp\Big(\mathrm{kl}\big(r(s,a) + \rho,\ r(s,a)\big)\Big)}{1 - \exp\Big(-\mathrm{kl}\big(r(s,a) + \rho,\ r(s,a)\big)\Big)} + \frac{\exp\Big(\mathrm{kl}\big(r(s,a) - \rho,\ r(s,a)\big)\Big)}{1 - \exp\Big(-\mathrm{kl}\big(r(s,a) - \rho,\ r(s,a)\big)\Big)}$$

$$+ \sum_{s'} \Bigg[ \frac{\exp\Big(\mathrm{kl}\big(p(s'|s,a) + \rho/S,\ p(s'|s,a)\big)\Big)}{1 - \exp\Big(-\mathrm{kl}\big(p(s'|s,a) + \rho/S,\ p(s'|s,a)\big)\Big)}$$

$$+ \frac{\exp\Big(\mathrm{kl}\big(p(s'|s,a) - \rho/S,\ p(s'|s,a)\big)\Big)}{1 - \exp\Big(-\mathrm{kl}\big(p(s'|s,a) - \rho/S,\ p(s'|s,a)\big)\Big)} \Bigg] \Bigg),$$

and using (49-53) we get:

$$\mathbb{P}(\overline{\mathcal{C}_T^1(\xi)} \cap \mathcal{E}_T) \leq \sum_{t=T^{1/4}}^{T} B \exp\left(-\frac{CT^{1/16}}{\sqrt{\log(1 + SAT^2)}}\right) \leq BT \exp\left(-\frac{CT^{1/16}}{\sqrt{\log(1 + SAT^2)}}\right).$$

Combined with (48), the previous inequality implies that:

$$\mathbb{P}\left(\overline{\mathcal{C}_T^1(\xi)}\right) \leq \frac{1}{T^2} + BT \exp\left(-\frac{CT^{1/16}}{\sqrt{\log(1 + SAT^2)}}\right).$$

$\square$

### F.4 Concentration of state-action visitation frequency

The following proposition is a somewhat stronger version of Proposition 12. The fact that $\widehat{\mathcal{M}}_t$ is within a distance of at most $\xi$ from $\mathcal{M}$ enables us to have a tighter control of the ergodicity constants $L_t$. This in turn allows us to derive a finite sample bound on the deviations of state-action frequency of visits. Before stating the result we recall some simple facts:

**Fact 1:** For any two policies $\pi, \pi'$ we have: $\|P_{\pi'} - P_\pi\|_\infty \leq \|\pi' - \pi\|_\infty$, where the norm is on policies viewed as vectors of $\mathbb{R}^{SA}$.

**Fact 2:** Under $\mathcal{C}_T^1(\xi)$, for $k \geq \sqrt{T}$ we have:

$$\left\|\overline{\pi_k^o} - \pi^o(\mathcal{M})\right\|_\infty \leq \frac{\sum_{j=1}^{T^{1/4}} \varepsilon_j \|\pi_u - \pi^o(\mathcal{M})\|_\infty}{k} + \frac{\sum_{j=T^{1/4}+1}^{k} \left\|\pi^o(\widehat{\mathcal{M}}_j) - \pi^o(\mathcal{M})\right\|_\infty}{k}$$

$$\leq \frac{T^{1/4}}{k} + \xi.$$

**Fact 3:** Under $\mathcal{C}_T^1(\xi)$, for $k \geq \sqrt{T}$ we have:

$$\|\omega_k - \omega^\star\|_1 \leq \kappa_\mathcal{M} \|P_k - P_{\pi^o}\|_\infty$$

$$\leq \kappa_\mathcal{M} \|\pi_k - \pi^o(\mathcal{M})\|_\infty$$

$$\leq \kappa_\mathcal{M} \big[\varepsilon_k + \left\|\overline{\pi_k^o} - \pi^o(\mathcal{M})\right\|_\infty\big]$$

$$\leq \kappa_\mathcal{M} \big[T^{\frac{-1}{4(m+1)}} + \frac{T^{1/4}}{k} + \xi\big].$$

where we used Lemma 22, Fact 1, the definitions of $\pi_k$ and $\varepsilon_k$ and Fact 2 respectively.

**Fact 4:** Under $\mathcal{C}_T^1(\xi)$, for $k \geq \sqrt{T}$ we have:

$$
\begin{aligned}
D(\pi_k, \pi_{k-1}) &= \|P_k - P_{k-1}\|_\infty \\
&\leq \|P_k - P_{\pi^\circ}\|_\infty + \|P_{\pi^\circ} - P_{k-1}\|_\infty \\
&\leq 2\big[T^{\frac{-1}{4(m+1)}} + \frac{T^{1/4}}{k-1} + \xi\big]
\end{aligned}
$$

**Proposition 19.** *Under C-Navigation, for all $\xi > 0$, there exists a time $T_\xi$ such that for all $T \geq T_\xi$, all $t \geq T^{3/4}$ and all functions $f : \mathcal{Z} \to \mathbb{R}^+$, we have:*

$$
\mathbb{P}\left(\left|\frac{\sum_{k=1}^t f(s_k, a_k)}{t} - \mathbb{E}_{(s,a)\sim\omega^\star}[f(s,a)]\right| \geq K_\xi \|f\|_\infty \xi \Big| \mathcal{C}_T^1(\xi)\right) \leq 2\exp\big(-t\xi^2\big),
$$

*where $\xi \mapsto K_\xi$ is a mapping with values in $(1,\infty)$ such that $\limsup_{\xi\to 0} K_\xi < \infty$. In particular, this implies that:*

$$
\mathbb{P}\left(\exists (s,a) \in \mathcal{Z}, \left|N_{sa}(t)/t - \omega_{sa}^\star\right| \geq K_\xi\xi \Big| \mathcal{C}_T^1(\xi)\right) \leq 2SA\exp\big(-t\xi^2\big).
$$

**Corollary 2.** *We have* $\mathbb{P}\left(\overline{\mathcal{C}_T^2(\xi)} \Big| \mathcal{C}_T^1(\xi)\right) \leq \dfrac{2SA\exp(-T^{3/4}\xi^2)}{1 - \exp(-\xi^2)}.$

*Proof.* Consider the difference

$$
\begin{aligned}
D &= \frac{\sum_{k=1}^t f(z_k)}{t} - \omega^\star(f) \\
&= \frac{\sum_{k=1}^{\sqrt{T}} [f(z_k) - \omega^\star(f)]}{t} + \frac{\sum_{k=\sqrt{T}+1}^t [f(z_k) - \omega^\star(f)]}{t} \\
&= \underbrace{\frac{\sum_{k=1}^{\sqrt{T}} [f(z_k) - \omega^\star(f)]}{t}}_{D_{1,t}} + \underbrace{\frac{\sum_{k=\sqrt{T}+1}^t \big[f(z_k) - \omega_{k-1}(f)\big]}{t}}_{D_{2,t}} + \underbrace{\frac{\sum_{k=\sqrt{T}+1}^t \big[\omega_{k-1}(f) - \omega^\star(f)\big]}{t}}_{D_{3,t}}.
\end{aligned}
$$
(54)

We clearly have:

$$
\forall T \geq \frac{1}{\xi^4}, \ \forall t \geq T^{3/4}, \ |D_{1,t}| \leq \frac{\|f\|_\infty \sqrt{T}}{t} \leq \frac{\|f\|_\infty}{T^{1/4}} \leq \|f\|_\infty \xi.
$$
(55)

Using Fact 3 and integral-series comparison, we upper bound the third term as follows:

$$\forall T \geq \left(\frac{2}{\xi}\right)^{4(m+1)}, \; \forall t \geq T^{3/4}, \; |D_{3,t}| \leq \kappa_{\mathcal{M}} \|f\|_{\infty} \frac{\sum_{k=\sqrt{T}+1}^{t-1} \|\omega_k - \omega^\star\|_1}{t}$$

$$\leq \kappa_{\mathcal{M}} \|f\|_{\infty} \frac{\sum_{k=\sqrt{T}+1}^{t-1} \left[T^{\frac{-1}{4(m+1)}} + \frac{T^{1/4}}{k} + \xi\right]}{t}$$

$$\leq \kappa_{\mathcal{M}} \|f\|_{\infty} \left(T^{\frac{-1}{4(m+1)}} + \frac{\sum_{k=\sqrt{T}+1}^{t-1} \frac{T^{1/4}}{k}}{t} + \xi\right)$$

$$\leq \kappa_{\mathcal{M}} \|f\|_{\infty} \left(T^{\frac{-1}{4(m+1)}} + \xi + \frac{T^{1/4} \log(t)}{t}\right)$$

$$\leq \kappa_{\mathcal{M}} \|f\|_{\infty} \left(T^{\frac{-1}{4(m+1)}} + \xi + \frac{T^{1/4}}{\sqrt{t}}\right)$$

$$\leq \kappa_{\mathcal{M}} \|f\|_{\infty} \left(T^{\frac{-1}{4(m+1)}} + \xi + T^{\frac{-1}{8}}\right)$$

$$\leq \kappa_{\mathcal{M}} \|f\|_{\infty} \left(2T^{\frac{-1}{4(m+1)}} + \xi\right)$$

$$\leq 2\kappa_{\mathcal{M}} \|f\|_{\infty} \xi. \tag{56}$$

Now to bound $D_{2,t}$ we use the function $\widehat{f}_k$ solution to the Poisson equation $\left(\widehat{f}_k - P_k \widehat{f}_k\right)(.) = f(.) - \omega_k(f)$. By Lemma 23, $\widehat{f}_k(.) = \sum_{n \geq 0} P_k^n[f - \omega_k(f)](.)$ exists and is solution to the Poisson equation. Therefore we can rewrite $D_{2,t}$ as follows:

$$D_{2,t} = \frac{\sum_{k=\sqrt{T}+1}^{t} \left[\widehat{f}_{k-1}(z_k) - P_{k-1}\widehat{f}_{k-1}(z_k)\right]}{t}$$

$$= M_t + C_t + R_t. \tag{57}$$

where

$$M_t \triangleq \frac{\sum_{k=\sqrt{T}+1}^{t} \left[\widehat{f}_{k-1}(z_k) - P_{k-1}\widehat{f}_{k-1}(z_{k-1})\right]}{t}.$$

$$C_t \triangleq \frac{\sum_{k=\sqrt{T}+1}^{t} \left[P_k \widehat{f}_k(z_k) - P_{k-1}\widehat{f}_{k-1}(z_k)\right]}{t}.$$

$$R_t \triangleq \frac{P_{\sqrt{T}}\widehat{f}_{\sqrt{T}}(z_{\sqrt{T}}) - P_t \widehat{f}_t(z_t)}{t}.$$

**Bounding $M_t$:** Note that $S_t \triangleq tM_t$ is a martingale since $\mathbb{E}[\widehat{f}_{k-1}(z_k)|\mathcal{F}_{k-1}] = P_{k-1}\widehat{f}_{k-1}(z_{k-1})$. Furthermore, by Lemma 23:

$$|S_k - S_{k-1}| = |\widehat{f}_{k-1}(z_k) - P_{k-1}\widehat{f}_{k-1}(z_{k-1})|$$

$$\leq 2\left\|\widehat{f}_{k-1}\right\|_{\infty}$$

$$\leq 2\|f\|_{\infty} L_{k-1}. \tag{58}$$

Recall from Lemma 13 that $L_k = \mathcal{L}(\varepsilon_k, \overline{\pi}_k^o, \omega_k)$ where:

$$\mathcal{L}(\varepsilon, \pi, \omega) \triangleq \frac{2}{\theta(\varepsilon, \pi, \omega)\left[1 - \theta(\varepsilon, \pi, \omega)^{1/r}\right]}$$

$$\theta(\varepsilon, \pi, \omega) \triangleq 1 - \sigma(\varepsilon, \pi, \omega).$$

$$\sigma(\varepsilon, \pi, \omega) \triangleq \left[\varepsilon^r + \left((1-\varepsilon)A \min_{s,a} \pi(a|s)\right)^r\right]\sigma_u\left(\min_z \frac{\omega_u(z)}{\omega(z)}\right).$$

Now for $T \geq \left(\frac{2}{\xi}\right)^{4(m+1)}$ and $k \geq \sqrt{T}$ we have:

$$|\varepsilon_k| = k^{\frac{-1}{2(m+1)}} \leq T^{\frac{-1}{4(m+1)}} \leq \xi/2.$$

$$\left\|\overline{\pi}_k^o - \pi^o(\mathcal{M})\right\|_\infty \leq \frac{T^{1/4}}{k} + \xi \leq \frac{1}{T^{1/4}} + \xi \leq 2\xi.$$

$$\|\omega_k - \omega^\star\|_1 \leq \kappa_\mathcal{M}\left[T^{\frac{-1}{4(m+1)}} + \frac{T^{1/4}}{k} + \xi\right] \leq \kappa_\mathcal{M}\left[2T^{\frac{-1}{4(m+1)}} + \xi\right] \leq 2\kappa_\mathcal{M}\xi.$$

Therefore:

$$L_k \leq L_\xi \triangleq \sup_{\substack{|\varepsilon| \leq \xi/2 \\ \|\pi - \pi^o(\mathcal{M})\|_\infty \leq 2\xi \\ \|\omega - \omega^\star\|_1 \leq 2\kappa_\mathcal{M}\xi}} \mathcal{L}(\varepsilon, \pi, \omega). \tag{59}$$

Using (58), (59) and Azuma-Hoeffding inequality we get for all $t \geq T^{3/4}$:

$$\begin{aligned}
\mathbb{P}\left(|M_t| \geq 2\|f\|_\infty L_\xi \xi\right) &= \mathbb{P}\left(|S_t| \geq 2t\|f\|_\infty L_\xi \xi\right) \\
&= \mathbb{P}\left(|S_t - S_{\sqrt{T}}| \geq 2t\|f\|_\infty L_\xi \xi\right) \\
&\leq 2\exp\left(\frac{-t^2\xi^2}{(t - \sqrt{T})}\right) \\
&\leq 2\exp\left(-t\xi^2\right). \tag{60}
\end{aligned}$$

**Bounding $C_t$:** Using Lemma 23 we have for all $T \geq \left(\frac{2}{\xi}\right)^{4(m+1)}$ and all $t \geq T^{3/4}$:

$$\begin{aligned}
|C_t| &\leq \|f\|_\infty \frac{\sum_{k=\sqrt{T}+1}^{t} L_k\left[\|\omega_k - \omega_{k-1}\|_1 + L_{k-1}D(\pi_k, \pi_{k-1})\right]}{t} \\
&\leq \|f\|_\infty \frac{\sum_{k=\sqrt{T}+1}^{t} L_\xi\left[\|\omega_k - \omega^\star\|_1 + \|\omega^\star - \omega_{k-1}\|_1 + 2L_\xi\left(T^{\frac{-1}{4(m+1)}} + \frac{T^{1/4}}{k-1} + \xi\right)\right]}{t} \\
&\leq \|f\|_\infty \frac{\sum_{k=\sqrt{T}+1}^{t} L_\xi\left[\kappa_\mathcal{M}\left(2T^{\frac{-1}{4(m+1)}} + 2\xi + \frac{T^{1/4}}{k} + \frac{T^{1/4}}{k-1}\right) + 2L_\xi\left(T^{\frac{-1}{4(m+1)}} + \frac{T^{1/4}}{k-1} + \xi\right)\right]}{t} \\
&\leq 2\|f\|_\infty (\kappa_\mathcal{M} L_\xi + L_\xi^2)\left[T^{\frac{-1}{4(m+1)}} + \xi + T^{1/4}\frac{\log(t)}{t}\right] \\
&\leq 2\|f\|_\infty (\kappa_\mathcal{M} L_\xi + L_\xi^2)\left[T^{\frac{-1}{4(m+1)}} + \xi + T^{1/4}\frac{1}{\sqrt{t}}\right] \\
&\leq 2\|f\|_\infty (\kappa_\mathcal{M} L_\xi + L_\xi^2)\left[T^{\frac{-1}{4(m+1)}} + \xi + T^{-1/8}\right] \\
&\leq 4\|f\|_\infty (\kappa_\mathcal{M} L_\xi + L_\xi^2)\xi, \tag{61}
\end{aligned}$$

where the second line comes from (59) and Fact 4 and the third line is due to Fact 3.

**Bounding $R_t$:**    Finally, by Lemma 23 we have:

$$\forall T \geq \left(\frac{2}{\xi}\right)^{4(m+1)}, \ \forall t \geq T^{3/4}, \ |R_t| \leq \frac{\left\|\widehat{f}_{\sqrt{T}}\right\|_\infty + \left\|\widehat{f}_t\right\|_\infty}{t}$$

$$\leq \frac{\|f\|_\infty \left(L_{\sqrt{T}} + L_t\right)}{t}$$

$$\leq 2 \|f\|_\infty L_\xi T^{-3/4}$$

$$\leq 2 \|f\|_\infty L_\xi \xi. \tag{62}$$

Summing up the inequalities (55-62) yields for all $T \geq \left(\frac{2}{\xi}\right)^{4(m+1)}$ and all $t \geq T^{3/4}$:

$$\mathbb{P}\left(\left|\frac{\sum_{k=1}^{t} f(z_k)}{t} - \omega^\star(f)\right| \geq K_\xi \|f\|_\infty \xi \Big| \mathcal{C}_T^1(\xi)\right) \leq 2 \exp\left(-t\xi^2\right). \tag{63}$$

where $K_\xi \triangleq 1 + 2\kappa_\mathcal{M} + 4L_\xi(1 + \kappa_\mathcal{M} + L_\xi)$. Note that $\limsup_{\xi \to 0} L_\xi = \mathcal{L}(0, \pi^o(\mathcal{M}), \omega^\star) < \infty$ [20] implying that $\limsup_{\xi \to 0} K_\xi < \infty$. We get the final result by applying (63) to indicator functions $\mathbb{1}_{s,a}(z)$ and using a union bound. □

# G   Technical Lemmas

## G.1   Upper bound on the norm of products of substochastic matrices

Before we proceed with the lemma, we lay out some definitions. $\eta_1 \triangleq \min\left\{P_{\pi_u}(z, z') \,\big|\, (z, z') \in \mathcal{Z}^2, P_{\pi_u}(z, z') > 0\right\}$ denotes the minimum positive probability of transition in $\mathcal{M}$. Similarly define $\eta_2 \triangleq \min\left\{P_{\pi_u}^n(z, z') \,\big|\, (z, z') \in \mathcal{Z}^2, n \in [[1, m+1]], P_{\pi_u}^n(z, z') > 0\right\}$ the minimal probability of reaching some state-action pair $z'$ from any other state-action $z$ after $n \leq m+1$ [21] transitions in the Markov chain induced by the uniform random policy. Finally, $\eta \triangleq \eta_1 \eta_2$.

**Lemma 20.** *Fix some state-action $z$ and let $P_t$ be the transition matrix under some policy $\pi_t$ satisfying $\pi_t(a|s) \geq \varepsilon_t \pi_u(a|s)$ for all $(s, a) \in \mathcal{Z}$. Define the substochastic matrix $Q_t$ obtained by removing from $P_t$ the row and the column corresponding to $z$:*

$$P_t = \left(\begin{array}{c|c} Q_t & [P_t(z', z)]_{z' \neq z} \\ \hline [P_t(z, z')]_{z' \neq z}^T & P_t(z, z) \end{array}\right).$$

*Then we have:*

$$\forall n \geq 1, \ \left\|\prod_{l=n+1}^{n+m+1} Q_l\right\|_\infty \leq 1 - \eta \prod_{l=n+1}^{n+m+1} \varepsilon_l.$$

*Proof.* Define $r_k(n_1, n_2) = \sum_{j=1}^{SA-1} \left(\prod_{l=n_1+1}^{n_2} Q_l\right)_{kj}$ the sum of the k-th row in the product of matrices $Q_l$ for $l \in [[n_1+1, n_2]]$. We will prove that or all $i \in [[1, SA-1]]$: $r_i(n, n+m+1) \leq 1 - \eta \prod_{l=n+1}^{n+m+1} \varepsilon_l$.

The result follows immediately by noting that $\left\|\prod_{l=n+1}^{n+m+1} Q_l\right\|_\infty = \max_{i \in [[1, SA-1]]} r_i(n, n+m+1)$.

---

[20]Refer to (34) and (35) for a formal justification.

[21]Refer to the preamble of Appendix D for more detail.

Consider $z'$ such that $P_{\pi_u}(z', z) \geq \eta_1$ (such $z'$ always exists since $\mathcal{M}$ is communicating) and let $k^\star$ be the index of the row corresponding to $z'$ in $Q_t$. Then for all $n_1 \geq 1$:

$$
\begin{aligned}
r_{k^\star}(n_1, l = n_1 + 1) &= \sum_{j=1}^{SA-1} (Q_{n_1+1})_{k^\star j} \\
&= 1 - P_{n_1+1}(z', z) \\
&\leq 1 - \eta_1 \varepsilon_{n_1+1}.
\end{aligned}
\tag{64}
$$

Now for $n_1, n_2 \geq 1$ we have:

$$
\begin{aligned}
r_{k^\star}(n_1, n_1 + n_2) &= \sum_{j_1=1}^{SA-1} \left( \prod_{l=n_1+1}^{n_1+n_2} Q_l \right)_{k^\star j_1} \\
&= \sum_{j_1=1}^{SA-1} \sum_{j_2=1}^{SA-1} \left( \prod_{l=n_1+1}^{n_1+n_2-1} Q_l \right)_{k^\star j_2} (Q_{n_1+n_2})_{j_2 j_1} \\
&= \sum_{j_2=1}^{SA-1} \left( \prod_{l=n_1+1}^{n_1+n_2-1} Q_l \right)_{k^\star j_2} \left[ \sum_{j_1=1}^{SA-1} (Q_{n_1+n_2})_{j_2 j_1} \right] \\
&= \sum_{j_2=1}^{SA-1} \left( \prod_{l=n_1+1}^{n_1+n_2-1} Q_l \right)_{k^\star j_2} r_{j_2}(n_1 + n_2 - 1, n_1 + n_2) \\
&\leq r_{k^\star}(n_1, n_1 + n_2 - 1) \\
&\;\; \vdots \\
&\leq r_{k^\star}(n_1, n_1 + 1) \\
&\leq 1 - \eta_1 \varepsilon_{n_1+1},
\end{aligned}
\tag{65}
$$

where in the fifth line we use the fact that for all $j_2, a, b$: $r_{j_2}(a, b) \leq 1$ since the matrices $Q_l$ are substochastic. The last line comes from (64). Now for all other indexes $i \in [|1, SA - 1|]$ we have:

$$
\begin{aligned}
\forall n_1 \in [|1, m|], \; r_i(n, n + m + 1) &= \sum_{j_1=1}^{SA-1} \left( \prod_{l=n+1}^{n+n_1} Q_l \times \prod_{l=n+n_1+1}^{n+m+1} Q_l \right)_{ij_1} \\
&= \sum_{j_1=1}^{SA-1} \sum_{j_2=1}^{SA-1} \left( \prod_{l=n+1}^{n+n_1} Q_l \right)_{ij_2} \left( \prod_{l=n+n_1+1}^{n+m+1} Q_l \right)_{j_2 j_1} \\
&= \sum_{j_2=1}^{SA-1} \left( \prod_{l=n+1}^{n+n_1} Q_l \right)_{ij_2} \sum_{j_1=1}^{SA-1} \left( \prod_{l=n+n_1+1}^{n+m+1} Q_l \right)_{j_2 j_1} \\
&= \sum_{j_2=1}^{SA-1} \left( \prod_{l=n+1}^{n+n_1} Q_l \right)_{ij_2} r_{j_2}(n + n_1, n + m + 1) \\
&\leq (1 - \eta_1 \varepsilon_{n+n_1+1}) \left( \prod_{l=n+1}^{n+n_1} Q_l \right)_{ik^\star} + \sum_{j_2 \neq k^\star} \left( \prod_{l=n+1}^{n+n_1} Q_l \right)_{ij_2} \\
&\leq (1 - \eta_1 \varepsilon_{n+n_1+1}) \left( \prod_{l=n+1}^{n+n_1} Q_l \right)_{ik^\star} + 1 - \left( \prod_{l=n+1}^{n+n_1} Q_l \right)_{ik^\star} \\
&= 1 - \eta_1 \varepsilon_{n+n_1+1} \left( \prod_{l=n+1}^{n+n_1} Q_l \right)_{ik^\star},
\end{aligned}
\tag{66}
$$

where we used (65) and the fact that the matrix $\prod_{l=n+1}^{n+n_1} Q_l$ is substochastic. Now since $\mathcal{M}$ is communicating then we can reach state-action $z'$ from any other state-action $z_i \in [|1, SA - 1|]$, after some

$n_i \leq m+1$ steps in the Markov chain corresponding to the random uniform policy. In other words, if $i$ is the index corresponding to $z_i$ then there exists $n_i \leq m+1$, such that $(P_{\pi_u}^{n_i})_{ik^\star} \geq \eta_2 > 0$. Therefore:

$$
\begin{aligned}
\left( \prod_{l=n+1}^{n+n_i} Q_l \right)_{ik^\star} &\geq \left( \prod_{l=n+1}^{n+n_i} \varepsilon_l P_{\pi_u} \right)_{ik^\star} \\
&= \left( \prod_{l=n+1}^{n+n_i} \varepsilon_l \right) (P_{\pi_u}^{n_i})_{ik^\star} \\
&\geq \eta_2 \prod_{l=n+1}^{n+n_i} \varepsilon_l.
\end{aligned}
\tag{67}
$$

Thus, combining (66) for $n_1 = n_i$ and (67) we get:

$$
\begin{aligned}
\forall i \in [\![1, SA-1]\!], \ r_i(n, n+m+1) &\leq 1 - \eta_1 \eta_2 \prod_{l=n+1}^{n+n_i} \varepsilon_l \\
&\leq 1 - \eta_1 \eta_2 \prod_{l=n+1}^{n+m+1} \varepsilon_l \\
&= 1 - \eta \prod_{l=n+1}^{n+m+1} \varepsilon_l.
\end{aligned}
$$

$\square$

### G.2 Geometric ergodicity: a general result

The following lemma is adapted from the proof of the Convergence theorem (Theorem 4.9, [LPW06]).

**Lemma 21.** *Let $P$ be a stochastic matrix with stationary distribution vector $\omega$. Suppose that there exist $\sigma > 0$ and an integer $r$ such that $P^r(s, s') \geq \sigma \omega(s')$ for all $(s, s')$. Let $W$ be a rank-one matrix whose rows are equal to $\omega^\intercal$. Then:*

$$
\forall n \geq 1, \ \|P^n - W\|_\infty \leq 2\theta^{\frac{n}{r}-1}
$$

*where $\theta = 1 - \sigma$.*

*Proof.* We write: $P^r = (1-\theta)W + \theta Q$ where $Q$ is a stochastic matrix. Note that $WP^k = W$ for all $k \geq 0$ since $\omega^\intercal = \omega^\intercal P$. Furthermore $MW = W$ for all stochastic matrices since all rows of $W$ are equal. Using these properties, we will show by induction that $P^{rk} = (1-\theta^k)W + \theta^k Q^k$:
For $k = 1$ the result is trivial. Now suppose that $P^{rk} = (1-\theta^k)W + \theta^k Q^k$. Then:

$$
\begin{aligned}
P^r(k+1) &= P^{rk}P^r \\
&= [(1-\theta^k)W + \theta^k Q^k]P^r \\
&= (1-\theta^k)WP^r + (1-\theta)\theta^k Q^k W + \theta^{k+1}Q^{k+1} \\
&= (1-\theta^k)W + 1-\theta)\theta^k W + \theta^{k+1}Q^{k+1} \\
&= (1-\theta^{k+1})W + \theta^{k+1}Q^{k+1}.
\end{aligned}
$$

Therefore the result holds for all $k \geq 1$. Therefore $P^{rk+j} - W = \theta^k(Q^k P^j - W)$ which implies:

$$
\begin{aligned}
\forall n = rk + j \geq 1, \ \|P^n - W\|_\infty &\leq \theta^k \|Q^k P^j - W\|_\infty \\
&\leq 2\theta^k = 2\theta^{\lfloor \frac{n}{r} \rfloor} \leq 2\theta^{\frac{n}{r}-1}.
\end{aligned}
$$

$\square$

### G.3 Condition number of Markov Chains

**Lemma 22.** *(Theorem 2 in [Sch68]) Let $P_1$ (resp. $P_2$) be the transition kernel of a Markov Chain with stationary distribution $\omega_1$ (resp. $\omega_2$). Define $Z_1 \triangleq (I - P_1 + \mathbb{1}\omega_1^\mathsf{T})^{-1}$. Then:*

$$\omega_2^\mathsf{T} - \omega_1^\mathsf{T} = \omega_2^\mathsf{T}[P_2 - P_1]Z_1 \quad and \quad \|\omega_2 - \omega_1\|_1 \leq \kappa_1 \|P_2 - P_1\|_\infty .$$

*where $\kappa_1 \triangleq \|Z_1\|_\infty$. Crucially, in our setting this implies that there exists a constant $\kappa_\mathcal{M}$ that only depends on $\mathcal{M}$ such that for all $\pi$:*

$$\|\omega_\pi - \omega^\star\|_1 \leq \kappa_\mathcal{M} \|P_\pi - P_{\pi^\circ}\|_\infty .$$

*where $\kappa_\mathcal{M} \triangleq \|Z_{\pi^\circ}\|_\infty = \left\|(I - P_{\pi^\circ} + \mathbb{1}\omega^{\star\mathsf{T}})^{-1}\right\|_\infty$.*

### G.4 Properties of Poisson equation's solutions

**Lemma 23.** *Let $P_\pi$ be a Markov transition kernel satisfying the assumptions (B1) and (B3) and denote by $\omega_\pi$ its stationary distribution. Then for any a bounded function $f : \mathcal{Z} \to \mathbb{R}^+$, the function defined by $\widehat{f}_\pi(.) \triangleq \sum_{n \geq 0} P_\pi^n[f - \omega_\pi(f)](.)$ is well defined and is solution to the Poisson equation $(\widehat{f}_\pi - P_\pi\widehat{f}_\pi)(.) = f(.) - \omega_\pi(f)$. Furthermore:*

$$\left\|\widehat{f}_\pi\right\|_\infty \leq L_\pi \|f\|_\infty ,$$

*and for any pair of kernels $P_\pi, P_{\pi'}$:*

$$\left\|P_{\pi'}\widehat{f}_{\pi'} - P_\pi\widehat{f}_\pi\right\|_\infty \leq L_{\pi'} \|f\|_\infty \left[\|\omega_{\pi'} - \omega_\pi\|_1 + L_\pi D(\pi', \pi)\right].$$

*Proof.* We will prove that $\widehat{f}_\pi$ is well defined. Checking that it satisfies the Poisson equation is straightforward. Observe that:

$$\widehat{f}_\pi(.) \triangleq \sum_{n \geq 0} P_\pi^n[f - \omega_\pi(f)](.)$$

$$= \sum_{n \geq 0} [P_\pi^n - W_\pi][f - \omega_\pi(f)](.),$$

where the second equality is because $(W_\pi f)(z) = \omega_\pi(f)$ for all $z \in \mathcal{Z}$. From the last expression, we see that the sum defining $\widehat{f}_\pi$ converges and we have the first bound $\left\|\widehat{f}_\pi\right\|_\infty \leq L_\pi \|f\|_\infty$. Now for the second bound, we write:

$$(P_{\pi'}\widehat{f}_{\pi'} - P_\pi\widehat{f}_\pi)(.) = \sum_{n \geq 1} \left[P_{\pi'}^n \left[\omega_\pi(f) - \omega_{\pi'}(f)\right] + \left[P_{\pi'}^n - P_\pi^n\right]\left[f - \omega_\pi(f)\right]\right](.)$$

$$= \underbrace{\sum_{n \geq 1} P_{\pi'}^n \left[\omega_\pi(f) - \omega_{\pi'}(f)\right](.)}_{A(.)} + \underbrace{\sum_{n \geq 1} \left[P_{\pi'}^n - P_\pi^n\right]\left[f - \omega_\pi(f)\right](.)}_{B(.)}. \quad (68)$$

Using the same trick as before we obtain:

$$\|A\|_\infty \leq \|f\|_\infty C_{\pi'}(1 - \rho_{\pi'})^{-1} \|\omega_\pi - \omega_{\pi'}\|_1$$
$$= \|f\|_\infty L_{\pi'} \|\omega_\pi - \omega_{\pi'}\|_1 . \quad (69)$$

On the other hand, a simple calculation shows that $(B - P_{\pi'}B)(.) = (P_{\pi'} - P_\pi)\widehat{f}_\pi(.)$, ie $B$ is solution to the modified Poisson equation where the right hand side is $(P_{\pi'} - P_\pi)\widehat{f}_\pi$. Therefore:

$$B(.) = \sum_{n \geq 0} P_{\pi'}^n\left[(P_{\pi'} - P_\pi)\widehat{f}_\pi\right](.).$$

and

$$\begin{aligned}
\|B\|_\infty &\leq L_{\pi'} \left\| (P_{\pi'} - P_\pi) \widehat{f}_\pi \right\|_\infty \\
&\leq L_{\pi'} D(\pi', \pi) \left\| \widehat{f}_\pi \right\|_\infty \\
&\leq L_\pi L_{\pi'} D(\pi', \pi) \|f\|_\infty .
\end{aligned} \tag{70}$$

Summing up equation (68) and inequalities (69-70) ends the proof. $\qquad\square$