# OpenReview forum: "Navigating to the Best Policy in Markov Decision Processes"
_NeurIPS.cc/2021/Conference — NeurIPS 2021 Poster_

### Official Review · Reviewer_1sx6 · 2021-07-02

**Rating:** 5
**Confidence:** 4

**Summary:**

The paper studies best-policy identification in discounted MDPs with online interaction. The authors first derive an information-theoretic lower bound on the sample complexity of any \delta-correct algorithm for this problem. The lower bound is given as the solution to a non-convex optimization problem for which a convex relaxation is provided. The authors then design an algorithm using the standard template for building pure exploration strategies from asymptotic lower bounds (GLR test + tracking + forced exploration). For this algorithm, the authors derive an asymptotic (as \delta -> 0) upper bound on the sample complexity that matches the value of the relaxed optimization problem up to a factor 2.

**Limitations And Societal Impact:**

The authors discuss the limitations of their algorithm and leave their solution to future work. I do not see any societal impact.

**Main Review:**

Significance, novelty, and relevance
---

The paper considers a relevant problem. I found this contribution interesting and significant. To my knowledge, this is the first work to provide problem-dependent sample complexity bounds for BPI in the online setting (i.e., without generative model). In particular, I appreciated the authors' efforts in describing the main complications that arise in the online setting (e.g., how to handle the navigation constraints).

Soundness of the claims
---

All claims seem sound, though I did not check all proofs in detail.


Relation with prior work
---

Relevant prior works are discussed. The paper could also mention recent works on asymptotically-optimal algorithms for pure-exploration/regret-minimization in bandits which remove the need for forced exploration and for oracles for computing the lower bound (see detailed comments below).


Clarity of writing
---

The first part of the paper is very well-written, while I found Sec. 4 quite hard to follow. I think the main issue is that the proposed algorithm is only briefly sketched in few lines of text, while no detailed pseudo-code is provided. This made it hard for me to understand what algorithm is actually analyzed (see also detailed comments below). Then, the section tries to explain the algorithm's components in detail, though, again, these are slightly difficult to understand since no reference pseudo-code is provided.


Detailed comments/questions
---

I have two major concerns about this work:

1. Only an asymptotic sample complexity bound is provided for the proposed algorithm. This, in my view, has two main limitations. (1) Having only an asymptotic result might be of limited significance since it might not describe well the behavior of an algorithm in the moderate-confidence setting, which is typically used in practice. (2) An asymptotic bound is typically quite "implicit". Hiding low-order (in log(1/delta)) terms makes it hard to understand the role played by each component in determining the final sample complexity. Please note that I am not advocating against asymptotic optimality. I think it is very good to have an algorithm that is asymptotically optimal, but I also think that the asymptotic result should be a special case of an explicit sample-complexity bound for the moderate-confidence setting which clearly shows the impact of each component.

2. The proposed algorithm uses forced exploration to guarantee a minimum level of estimation accuracy for the true underlying MDP. While forced exploration is a simple trick to guarantee asymptotic optimality for this kind of algorithms, even in the simpler MAB case it is known that when facing "hard instances" in the moderate-confidence setting the actual sample complexity observed in practice might be significantly affected (if not dominated) by forced-exploration (I think [1] provides examples of this). I believe this might be even amplified in MDPs, where there might be states that are very hard to reach with a uniform policy (which I guess partially explains the very slow theoretical rate for decaying forced exploration). Recent papers on building algorithms from lower bounds in the bandit literature removed the need for forced exploration by using confidence-intervals/optimism, see e.g. [1,2,3] for pure exploration and [4,5,6] for regret minimization. I think the proposed algorithm would significantly benefit from adopting such techniques.

Other comments:

3. I believe the assumption of unique optimal policy might be quite restrictive in the MDP setting. Is the assumption needed only for the analysis of the algorithm or also for the derivation of the lower bound? Any intuition on how to remove it?

4. The last paragraph of Sec. 3 mentions that a projected gradient-descent algorithm is used to compute the optimal allocation, while from Sec. 4 it seems that the algorithm has access to an oracle providing the optimal allocation for the empirical MDP at each time step. Which one is used in the algorithm and in its analysis?

5. While the algorithm design and analysis for the online setting is novel, it is also true that the paper builds on top of the recent work [7]. It would be good to clearly describe, at a technical level, how the two contributions differ (e.g., what components of the algorithm are similar/different, where the analysis is similar/different, etc.)

6. What policy does the \mu_min of Corollary 1 refer to? Also, do you think that the minimum state-action visitation probability of some policy really affects the sample complexity or that might be an artifact of the upper bound derived there and in Lemma 5?

7. Why is the sample complexity derived in Theorem 8 optimal only up to a factor of 2? Is it because two concentration inequalities are used in the analysis of the GLR test (one for p and one for q)? Any idea how to reduce that factor to 1?

Minor:

- Line 58: MDPs -> MDP
- Line 95: Acronym \delta-PC not introduced
- Line 114: do you need also absolute continuity of the reward kernels?
- Line 211: remove "a"
- Eq.(13): is there no dependence on \xi in the rhs?

[1] Degenne, Rémy, Wouter M. Koolen, and Pierre Ménard. "Non-asymptotic pure exploration by solving games." NeurIPS (2019).
[2] Degenne, Rémy, et al. "Gamification of pure exploration for linear bandits." International Conference on Machine Learning. PMLR, 2020.
[3] Zaki, Mohammadi, Avi Mohan, and Aditya Gopalan. "Explicit best arm identification in linear bandits using no-regret learners." arXiv preprint arXiv:2006.07562 (2020).
[4] Degenne, Rémy, Han Shao, and Wouter Koolen. "Structure adaptive algorithms for stochastic bandits." International Conference on Machine Learning. PMLR, 2020.
[5] Tirinzoni, Andrea, et al. "An asymptotically optimal primal-dual incremental algorithm for contextual linear bandits." NeurIPS (2020).
[6] Kirschner, Johannes, et al. "Asymptotically Optimal Information-Directed Sampling." arXiv preprint arXiv:2011.05944 (2020).
[7] Marjani, Aymen Al, and Alexandre Proutiere. "Adaptive sampling for best policy identification in Markov decision processes." arXiv preprint arXiv:2009.13405 (2020).

**Time Spent Reviewing:**

4

---

> ### Author Response · Authors · 2021-08-08
> **On asymptotic results and other remarks.**
>
> 1- We agree with the reviewer that ideally one would like to derive non-asymptortic and fully interpretable sample complexity results. We are not sure whether this ambitious objective has been achieved even in the simple case of bandit problems, despite the recent interesting cited papers. The case of MDPs is much more difficult – note that before this submission, the only papers providing a problem-specific analysis of the sample complexity are [Brunskill] and [7], but in the simple generative model case. Our paper is the first to propose a problem-dependent analysis for the online setting. Our main motivation in this work was to understand how the online learning scheme affects the sample complexity compared to the easier case where we have a generative model. A first step is to understand how the first order term T*log(1/delta) changes, which explains the focus on asymptotic bounds. We stress that proving convergence of visit-frequencies to the oracle allocation already involves several technical results about non-homogeneous markov chains (with history dependent kernels). Performing non-asymptotic analysis would require additional technical results about the mixing times of Markov chains induced by policies, at the risk of overloading the paper.
> - Finally, while asymptotic results can be of limited relevance, the same holds, mutatis mutandis, for Lai & Robbins' (L&R) lower bound in the context of regret minimization: it is also asymptotic, and also valid only in the long run while moderate horizon results would be preferable. However, this was obviously not a valid argument to disqualify L&R's bound, the consideration of which generated a large amount of theoretical and practical advances.
>
> 2- The reviewer mentions using no-regret learners to perform exploration. While this approach might be smarter than exploration using the uniform policy, it assumes that we have access to an oracle that solves the best response problem (ie finds the optimal alternative MDP). This oracle is then used by the MIN player of a two-player min-max game. Unfortunately, in contrast with bandits, the best response problem is difficult to solve for MDPs (see  [7]). (This is actually what motivates the use of the upper bound of Lemma 5). We believe that without additional assumptions on the structure of the MDP, exploration using the uniform policy with a decaying rate is a reasonable solution.
>
> 3- The assumption of  a unique optimal policy is both used in the derivation of the lower bound (It simplifies the form of the set of alternative MDPs) and the analysis of the algorithm. If one were to remove this assumption, we believe that a more relevant objective would be epsilon-optimal policy identification, where multiple correct answers can be returned. A lower bound for this objective can be derived in a straightforward fashion, similarly to previous works in multi-armed bandits [8,9]. However, we expect the optimization program of this new lower bound to be non-convex as well. The challenge then is to study the "new" set of alternative MDPs (where one epsilon-optimal policy becomes sub-optimal) to derive an upper bound similar to the bound of Lemma 5 (with the right dependence in epsilon and 1-gamma). The algorithm design for this new objective follows immediately using the same machinery.
>
>
> 4- In the analysis of the algorithm, we assume having an oracle that solves the optimization problem (8) ( line 162). In the implementation, we use projected gradient-descent with a fixed number of iterations.
>
> 5- [7] derived an upper bound of the best response problem of the lower bound, then used ideas from the classical track-and-stop algorithm to design an algorithm for BPI with a generative model. (When it comes to sampling, an MDP with a generative model is equivalent to a bandit with S\times A arms). This work adapts the change-of-measure lower bound from [7] to include navigation constraints on the set of allocation vectors. More importantly, it proposes novel sampling rules for this restricted sampling scheme, where only actions can be chosen by the agent and states are dictated by the MDP. The main challenge lies in proving that, under these sampling rules, the state-action visits converge to the desired allocation vector.
>
> 6- The purpose of Corollary 1 is to show that minimax bounds in the literature (see lines 174-178) can be recovered from our problem-specific bounds. The \mu_min refers to the minimum state-action occupancy vector of any fixed arbitrary policy that the agent would choose for exploration. The maximum (over the set of policies) of \mu_min describes how well policies can cover the state-actions. Therefore, we believe that it might affect the sample complexity, although in a loose manner since it doesn't make any distinction between state-action pairs (eg some might have smaller gaps and therefore need to be visited more frequently).
>
> 7- The factor of 2 indeed arises from the union boundof two distinct deviation inequalities. It can be removed by deriving a deviation inequality of the full distribution of rewards and transitions (p,q).
>
> [8] Degenne, Rémy and Wouter M. Koolen "Pure Exploration with multiple correct answers" (NeurIPS 2019)
> [9] A. Garivier and Emilie Kaufmann. Non-asymptotic sequential tests for overlapping hypotheses and application to near optimal arm identification in bandit models. arXiv preprint, Statistics Theory, arXiv:1905.03495, 2019.

---

> > ### Comment · Reviewer_1sx6 · 2021-08-20
> > **Follow-up**
> >
> > Dear authors,
> >
> > Thank you for your detailed response.
> >
> > Regarding the asymptotic results, I agree that focusing on the term multiplying log(1/\delta) is a good first step towards deriving problem-dependent sample-complexity bounds in the online setting. I would suggest the authors explicitly state this focus on the asymptotic regime in the paper. Moreover, if possible, it would still be good to provide some idea about how other relevant lower-order terms scale (e.g., the sample complexity introduced by forced exploration, which can be partially extracted by inspecting the proofs).
> >
> > Regarding the usage of no-regret learners, the authors mentioned that these are difficult to use in this context because computing the best response is difficult in the MDP setting. However, could we apply no-regret learners to optimize for the upper bound U(M,\omega) instead of the original lower bound (after all it is still a min-max problem)? Anyway, that would not change the asymptotic performance and, given that we are only concerned about that, I agree this could be treated as an orthogonal direction.

---

> > > ### Author Response · Authors · 2021-08-23
> > > **Answer**
> > >
> > >
> > > We thank the reviewer for now agreeing that focusing on the term multiplying log(1/\delta) is a good first step towards deriving problem-dependent sample-complexity bounds in the online setting. We will make sure to clearly convey this message in the paper.
> > >
> > > We have analyzed the proofs to extract the non-leading terms of the sample complexity of our algorithm. For any small enough \xi, the non-asypmtotic term will be \bigO( (2/\xi)^(4 \times(m+1)) + S^2 \times A ) where m is a communication parameter of the MDP( defined in lemma 6). It can range from m = 1 for MDPs where all state are connected by some transition to m = S-1 in the worst-case of a chain MDP (eg RIVERSWIM):
> > >
> > >  - The \bigO(S^2 \times A) term comes from the failure probabilities of concentration of rewards and transitions of the empirical MDP (see lemma 18 and its proof, line 770).
> > >
> > >  - The \bigO( (2/\xi)^(4 \times(m+1)) ) term comes our deviation inequality of state-action visit frequencies, which is valid only after a threshold time T_\xi (see Proposition 19 and its proof, line 807).
> > >
> > >  - Finally, the asymptotic term of the sample complexity would then be 2U_o(\mcal,\xi)\log(1/\delta) where U_o(\mcal,\xi) is the supremum of our upper bound in the neighborhood of \mcal (a ball of radius \xi in a certain norm).
> > >
> > > After these modifications (thank you very much for your help!), we believe that we have a paper with strong contributions, and the first work providing problem-specific results for the sample complexity of learning optimal policies in MDPs without a generative model. From the reviews, we feel that we addressed all the concerns / questions; please let us know if we missed something. We hope that the contributions of the paper can be acknowledged with better scores. Many thanks.

---

### Official Review · Reviewer_rSqS · 2021-07-14

**Rating:** 6
**Confidence:** 1

**Summary:**

This work talks about best policy identification problem in discounted infinite MDP. Instead of assuming a generator, this work is on the online setting of MDP, where the agent can only collect state action pairs through online interaction with the environment. In theory, the work provides a lower bound of the BPI problem. Then the work provides a model-based algorithm MDP-NaS, which is shown to converge to the optimal policy.

**Limitations And Societal Impact:**

Yes.

**Main Review:**

### originality
The paper is original for discussing BPI.

### quality
The paper is generally of good quality for having consistent setting and explanations.

### clarity
The paper is well written and easy to follow.

### significance
The paper is significant on the theory side, but I am not quite sure how the results can be applied.

### minor issues

1. Can the author also compare with Dong, Kefan, et al. ?

2. typo: line 80 \gamma \in (0, 1]

Dong, Kefan, et al. "Q-learning with ucb exploration is sample efficient for infinite-horizon mdp." arXiv preprint arXiv:1901.09311 (2019).


**Time Spent Reviewing:**

1

---

### Official Review · Reviewer_ewHn · 2021-07-16

**Rating:** 4
**Confidence:** 3

**Summary:**

This paper deals with online learning of policies for MDPs. Its main novelty is considering the "navigation constraints" related to this online setting where, in contrast with an episodic setting, the system must learn the policy in a single episode without being able to reset to an initial state. Bounds on sample complexity are derived and an algorithm for learning the best policy by exploiting the lower bounds is proposed.


**Limitations And Societal Impact:**

A brief discussion on limitation is provided in te Conclusions. Societal Impacts are not mentioned.

**Main Review:**

The paper is heavily theoretical. I find the topic interesting and relevant but had difficulties following the results. Part of this that this style of PAC analysis is not my area of expertise, but I think the authors could have been a bit clearer in their writing.

On a high level, they could motivate the need to deal with an online setting (rather than a generative/episodic setting a bit better). In particular, assuming an online setting will typically mean the agent will be interacting with the real environment. In such a setting, it seems to be that it is crucial to ensure safety of the learning process, and this paper doesn't discuss that at all. For example, I think works like [1] could be mentioned and a more clear motivation about the setting and problem would be worth it here.

In terms of the technical presentation, I think there's a lot of room for improvement, the paper assumes a whole lot of background knowledge. Some points that could be clarified:

* Is the term "navigation constraints" used in the literature? It is used in the abstract but not defined, and I could only understand what it meant after reading the paper further and understanding the setup. I think its meaning should be spelled out to the reader earlier.

* PAC acronym is undefined

* The authors mention the minimax framework but don't say what it is

* What is the difference between a generative setting and an episodic one? I don't find the term generative very intuitive...

* I don't think sigma in line 92 is defined

* I don't get the condition in line 114 for M << M'. If p(.|s,a) and p'(.|s,a) are two distributions over S, what does it mean for one to be << the other?

Minor:
Line 237 uses "pretty slowly", which is pretty informal for a research paper.


[1] Turchetta, M., Berkenkamp, F. and Krause, A., 2016. Safe exploration in finite markov decision processes with gaussian processes. Advances in Neural Information Processing Systems, 29, pp.4312-4320.

POST RESPONSE
After reading the other reviews and rebuttals, my feeling is that the paper needs a bit more work to iron out some kinks and make its main messages clearer. I encourage the authors to do so.

**Time Spent Reviewing:**

3

---

> ### Author Response · Authors · 2021-08-08
> **On Safety and nomenclature**
>
> 1-  We agree that practical RL algorithms in the online setting need to include safety in their objectives. However, our main motivation in this work was to understand how the online learning scheme affects the sample complexity compared to the easier case where we have a generative model (aka a simulator). For this purpose, we considered an "idealized" setting where the only concern is sample efficiency and proposed the first problem-specific analysis of this problem. Dealing with safety issues adds another layer of difficulty to the problem and requires a paper of its own.
>
> 2- The term "generative model" is widely used in the RL literature to design a simulator of the MDP, see [1] and references therein.
>
> Thanks for the comments on the presentation, we will make corrections in the final version accordingly.
>
> [1] Andrea Zanette, Mykel J Kochenderfer, and Emma Brunskill. "Almost horizon-free structure-aware best policy identification with a generative model" (NeurIPS 2019)

---

### Official Review · Reviewer_yb3J · 2021-07-17

**Rating:** 6
**Confidence:** 3

**Summary:**

The paper studies the best policy identification (BPI) problem in Markov Decision Processes, under navigation constraints, i.e., when accessing a single trajectory from the environment, as opposed to the generative model setting. The paper focuses first on a lower bound on the sample complexity obtained by extending existing results. Then, an algorithm is introduced, and the corresponding sample complexity is analyzed. The contribution is mainly theoretical.

**Limitations And Societal Impact:**

This is a theoretical paper, I do not forsee any societal impact.

**Main Review:**

- (Organization) I had some troubles in reading the appendix, particularly when looking for the proofs of the results presented in the main paper. I couldn't find the proof of Proposition 4. I understand that it is a quite direct extension of Proposition 1 of [1], but I suggest stating it clearly, to avoid confusion. Moreover, I couldn't find the proof of Corollary 1 and Proposition 7.
- (About the upper bound to the lower bound sample complexity) The authors derive the lower bound to the sample complexity in Proposition 3, whose computation requires the solution of a non-convex optimization problem. Lemma 5 provides an upper bound to the sample complexity of Proposition 5, that can be solved with convex optimization. Therefore, I believe that the sample complexity of Lemma 5 is *not* a lower bound to the sample complexity of the problem. I suggest stating it clearly, because, in some sentences, I perceived some confusion. For instance, in the abstract, the authors refer to "relaxed lower bound" whose meaning is unclear to me.
- I read the appendix only partially

***Minor***
- line 80: ([0,1) - > [0,1)
- line 92: (ii) A - > (ii) a
- Line 186: Shouldn't that be an argmin?
- Line 195: too many closed paranthesis


***Overall***
The paper represents an interesting contribution to the theoretical understanding of the BPI problem in a more realistic setting compared with the generative model setting. The derivation of the lower bound seems a quite direct extension of [1]. Instead, the algorithm is novel. Therefore, I am inclined towards acceptance.


[1] Al Marjani, Aymen, and Alexandre Proutiere. "Adaptive sampling for best policy identification in Markov decision processes." In International Conference on Machine Learning, pp. 7459-7468. PMLR, 2021.


***Post Discussion***
I thank the authors for the feedback. I have read the reviews and the authors' responses. I still think that the paper addresses a relevant problem and that the proposed solution and corresponding analysis are interesting. I agree on the limitations about the asymptotic-regime bound and the forced exploration, although I think they are not a decisive factor. Nevertheless, I agree that the paper needs a quite deep revision to highlight the contributions, technical novelties, and results.

**Time Spent Reviewing:**

3

---

> ### Author Response · Authors · 2021-08-10
> **Organization and proofs**
>
> Thank you for your insightful comments. We will consider them and update our manuscript accordingly. Please answers to your questions below.
>
> Main review:
>
> 1. Proofs and organization. Proposition 4 is a direct consequence of Proposition 3, and its justification is given in the paragraph before we state the proposition. We will add a complete formal proof in the appendix. We will add a proof for Corollary 1 – it is explained in Appendix C2 and is similar to a result proved in [MP20]. Thanks for noticing. The proof of Proposition 7 is provided in Appendix E3 – but we agree that it is not easy to find, and will clarify. We will add appropriate pointers in the main text of the paper towards the proofs provided in Appendix.
>
> 2. About the upper bound to the lower bound sample complexity. Indeed, we derive an upper bound of the lower bound, and this new ‘relaxed’ bound is not a lower bound anymore. We believe the lower bound in Proposition 4 is tight, and hence it is important that our ‘relaxed’ bound is above the lower bound, so that we can devise an algorithm achieving a sample complexity corresponding to the relaxed bound. We will clarify as you suggest.
>
> Thanks for noticing typos, we will correct them.

---

> > ### Comment · Reviewer_yb3J · 2021-08-28
> > **Re: Organization and proofs**
> >
> > ***Post Discussion***
> > I thank the authors for the feedback. I have read the reviews and the authors' responses. I still think that the paper addresses a relevant problem and that the proposed solution and corresponding analysis are interesting. I agree on the limitations about the asymptotic-regime bound and the forced exploration, although I think they are not a decisive factor. Nevertheless, I agree that the paper needs a quite deep revision to highlight the contributions, technical novelties, and results.

---

### Official Review · Reviewer_Vy8C · 2021-08-15

**Rating:** 6
**Confidence:** 3

**Summary:**

The paper studies problem-dependent sample complexity bounds for best policy identification (BPI) in finite MDPs under the infinite horizon discounted criterion. The agent interacts sequentially (online) with the MDP, a weaker assumption than the generative model required by previous work in the same setting.
Its main contributions are:

(i)  A problem-dependent lower bound on the sample complexity that takes into account navigation constraints;

(ii) Since the lower bound is defined as the solution to a non-convex optimization problem, the paper uses a convex relaxation introduced by [MP20] to derive a relaxed lower bound;

(iii) An algorithm whose upper bound matches the relaxed lower bound up to a factor 2.


**Limitations And Societal Impact:**

Since this is a theoretical work, I believe there are no societal impacts to be addressed.


**Main Review:**


The paper studies a significant problem in reinforcement learning: the one of deriving sample complexity bounds and algorithms for best policy identification that work in an online setting and that are able to adapt to the structure of the MDP. It extends the results of [MP20] to the online setting.

With respect to the assumptions and the algorithm, I found the limitations below:

(1)  It is assumed that the MDP has a single optimal policy, which is quite strong.

(2)  The proposed algorithm relies on forced exploration based on a mixture with a uniformly random policy, and knowledge about the MDP structure is required to determine the optimal exploration rate. Without such knowledge, we must either use a very conservative rate, or rely on heuristics, or make the stronger assumption that the MDP is ergodic to use a minimal exploration rate.

(3)  Replacing the non-convex optimization problem by a convex relaxation is a very interesting approach, however, it leaves open the question of whether the “true” lower bound can be matched. If we assume that we have an oracle that gives the solution to the non-convex problem, would the same algorithm match the lower bound?  If so, it might be nice to add this remark in the paper.

Also, I believe that the paper has some points to be improved with respect to the presentation and discussion:

(i) It would be interesting to include a more detailed discussion about what are the main theoretical and algorithmic challenges that arise in the online setting, when compared to the generative setting studied by [MP20]. For instance, the lower bound seems to follow almost directly from the one in [MP20] and I think the challenges to derive the algorithm could be better explained.

(ii) It would be helpful to include a short description of the algorithm. In the current version, the description of the algorithm is mixed with theoretical results and discussion in Section 4.

(iii) In the experiments (Table 2), it seems that the empirical sample complexity of MDP-NaS is compared to an upper bound on the sample complexity of VRQL. However, it is possible that VRQL achieves the optimal policy with fewer samples than what is stated by the upper bound. I understand that this was done to compensate for the fact that VRQL has no stopping rule, but it would be interesting to see how many samples are actually required by VRQL to achieve the optimal policy in the MDPs that were tested. It might also be interesting to compare the algorithm to other BPI algorithms such as BPI-UCRL [KMDD+21].


**Time Spent Reviewing:**

7

---

> ### Author Response · Authors · 2021-08-15
> **Answer**
>
> (1) The assumption of unique optimal policy is both used in the derivation of the lower bound (It simplifies the form of the set of alternative MDPs) and the analysis of the algorithm . If one were to remove this assumption, we believe that a more relevant objective would be epsilon-optimal policy identification. A lower bound for this objective can be derived in a straightforward fashion, similarly to previous works in multi-armed bandits [1,2]. However, we expect the optimization program of this new lower bound to be non-convex as well. The challenge then is to study the "new" set of alternative MDPs (where one epsilon-optimal policy becomes sub-optimal) to derive an upper bound similar to the bound of Lemma 5 (with the right dependence in epsilon and 1-gamma). The algorithm design for this new objective follows immediately using the same machinery.
>
> (2) As far as we are aware, the only sampling rules that achieve asymptotic optimality for Pure Exploration in multi-armed bandits either use forced exploration or game-based exploration. The latter implements no-regret algorithms to simulate a two-player min-max game whose value is the inverse of the lower bound [3]. While this approach might be smarter than exploration using the uniform policy, it assumes that we have access to an oracle that solves the best response problem (ie finds the optimal alternative MDP). This oracle is then used by the MIN player of the min-max game. Unfortunately, in contrast with bandits, the best response problem is difficult to solve for MDPs (see [7]). (This is actually what motivates the use of the upper bound of Lemma 5). We believe that without additional assumptions on the structure of the MDP, exploration using the uniform policy with a decaying rate is a reasonable solution to achieve asymptotic optimality. Now indeed the appropriate decay rate involves the parameter m that depends on the MDP. However the algorithm only requires a reasonable upper bound on m. Note also that in the worst case, as illustrated in the example presented after Remark 1, m is close to S. If the MDP is ergodic (which is investigated most often in the literature), we do not have any problem choosing the exploration rate, which can be close as to 1/t as one wishes (see line 256). In our opinion, the choice of exploration rate in communicating MDPs clearly illustrates the challenges of navigating to learn in such MDPs.
>
> (3) The answer to this question depends on the nature of the set of solutions to the non-convex problem of the lower bound. This set is convex since the objective function is concave. If the set of optimal allocations is made of singleton then it is straightforward that the algorithm would match the lower bound. Otherwise, it would require some additional work to adapt Proposition 12 and show that the vector of visit-frequencies N(t)/t converges (in distance from its projection) to the set of optimal allocations. This was briefly mentioned in footnote 4. We will dedicate a separate remark for this statement in the final version of the paper.
>
> (i)- Indeed, except for the navigation constraints, the lower bound follows directly from [MP20]. However there are several technical novelties in the algorithm and the sampling rule in particular:
>     1) Unlike the generative model or bandit cases, forced exploration results in our setting become probabilistic. The challenge is then to prove a lower bound w.h.p on the state-action visits that has a good scaling w.r.t the failure probability \alpha (see corollary 2 in the appendix). This is crucial for asymptotic optimality, as the sum of probabilities of the failure events (where some state-action is visited less than ~T^(1/4) for some time step T) needs to be finite and independent of \delta (the failure probability of the algorithm).
>
>    2) To prove asymptotic optimality, we derived a new non-asymptotic analogue of the ergodic theorems for non-homogeneous markov chains (Proposition 10) to better control finite-time deviations of state-action visit-frequencies.
>
> (ii) Thanks for this remark. We will shorten the lower bound section and add a pseudo-code description of the algorithm in the final version.
>
> (iii) We will include the experiment with VRQL's pseudo-sample-complexity (without a stopping rule) in the final version. However, as VRQL relies on a fixed sampling rule that doesn't adapt to the MDP, we expect its pseudo-sample-complexity to still be larger than MDP-NaS sample complexity.
> As for BPI-UCRL, it was designed for the fixed horizon setting. In particular, BPI-UCRL relies on a backward recursion over time steps h \in [H], with Q function estimates initialized at zero for h=H. It is not clear how one can adapt the number of recursion steps and the initialization of Q-function estimates to the infinite-horizon discounted setting.
>
> [1] Degenne, Rémy and Wouter M. Koolen "Pure Exploration with multiple correct answers" (NeurIPS 2019)
> [2] A. Garivier and Emilie Kaufmann. Non-asymptotic sequential tests for overlapping hypotheses and application to near optimal arm identification in bandit models. arXiv preprint, Statistics Theory, arXiv:1905.03495, 2019.
> [3] Degenne, Rémy, Wouter M. Koolen, and Pierre Ménard. "Non-asymptotic pure exploration by solving games." NeurIPS (2019).

---

> > ### Comment · Reviewer_Vy8C · 2021-08-25
> > **A minor comment about the experiments**
> >
> > Thank you for your detailed response!
> >
> > Indeed, since VRQL does not adapt to the MDP, it's pseudo-sample-complexity should be larger. Regarding BPI-UCRL, it might be possible to set the horizon $H$ such that $\gamma^{H+1} / (1-\gamma) \leq \epsilon \leq \Delta_{\mathrm{min}}$ and use the Q function at $h=1$ to derive a policy.
> > In any case, my suggestion would be to investigate a little further (empirically) how adaptive the algorithm is, and if it performs well in finite-time.

---

> > > ### Author Response · Authors · 2021-08-27
> > > **Thanks**
> > >
> > > Thank you for your comments, we will include the additonnal experiments in the final version.

---

### Decision · Program_Chairs · 2021-09-27

**Decision:**

Accept (Poster)

**Comment:**

Although the scores for this paper make it a borderline case, after the discussion it seems that the only remaining issue is that the paper does not highlight better the contributions, technical novelties, and results. Further, the description of the algorithm was considered to be improvable. I think that these points should be manageable when preparing the final version, so that I recommend to accept the paper. The authors are expected to revise the paper taking into account the reviewers's comments and in particular paying attention to the points mentioned above.